# Bridging Information Asymmetry in Text-video Retrieval: A Data-centric Approach

**Zechen Bai**[1], **Tianjun Xiao**[2], **Tong He**[2], **Pichao Wang**[2], **Zheng Zhang**[2],
**Thomas Brox**[2,3], **Mike Zheng Shou**[1] *
[1]Show Lab, National University of Singapore    [2]Amazon    [3]University of Freiburg

## Abstract

As online video content rapidly grows, the task of text-video retrieval (TVR) becomes increasingly important. A key challenge in TVR is the information asymmetry between video and text: videos are inherently richer in information, while their textual descriptions often capture only fragments of this complexity. This paper introduces a novel, data-centric framework to bridge this gap by enriching textual representations to better match the richness of video content. During training, videos are segmented into event-level clips and captioned to ensure comprehensive coverage. During retrieval, a large language model (LLM) generates semantically diverse queries to capture a broader range of possible matches. To enhance retrieval efficiency, we propose a query selection mechanism that identifies the most relevant and diverse queries, reducing computational cost while improving accuracy. Our method achieves state-of-the-art results across multiple benchmarks, demonstrating the power of data-centric approaches in addressing information asymmetry in TVR. This work paves the way for new research focused on leveraging data to improve cross-modal retrieval.

## 1 Introduction

The explosion of online video content has generated an unprecedented demand for efficient and accurate text-video retrieval systems. As a key task in vision-language learning, text-video retrieval enables users to find semantically relevant videos based on natural language queries, making it indispensable for video search and recommendation systems. Typically, this task is approached by encoding both text and video into a joint latent space and calculating their similarity using a metric like cosine similarity. Yet, despite advances in the field, text-video retrieval remains fundamentally constrained by an often-overlooked problem: information asymmetry.

Information asymmetry refers to the inherent imbalance between the rich, multi-layered content of videos and their more concise textual descriptions. While videos can capture visual scenes, actions, and interactions in great detail, their associated text queries or captions often capture only fragments of this complexity. For example, in datasets like MSR-VTT (Xu et al., 2016b), some key moments in a video may be under-described or even omitted altogether, as illustrated in Fig. 1. This asymmetry complicates both the training and inference processes, as retrieval models may learn to focus on only specific parts of the video while disregarding other potentially relevant aspects. Moreover, when users issue text queries that fail to capture the full semantics of a target video, retrieval accuracy diminishes. These challenges raise an important question: can we mitigate information asymmetry by enriching textual representations to better match the richness of video content?

Most state-of-the-art approaches to text-video retrieval (Gorti et al., 2022; Wu et al., 2023b; Xue et al., 2022), especially those built on the CLIP architecture (Radford et al., 2021), assume a symmetrical relationship between text and video by projecting both into a shared latent space. However, this assumption fails to account for the vast disparity in information density between these two modalities. Some recent methods, such as T-MASS (Wang et al., 2024a) and UAVTR (Fang et al., 2023), have sought to address this imbalance by enriching text embeddings through stochastic or distribution-based mechanisms, yet they focus on model-centric solutions and largely overlook the underlying

---

*Corresponding Author

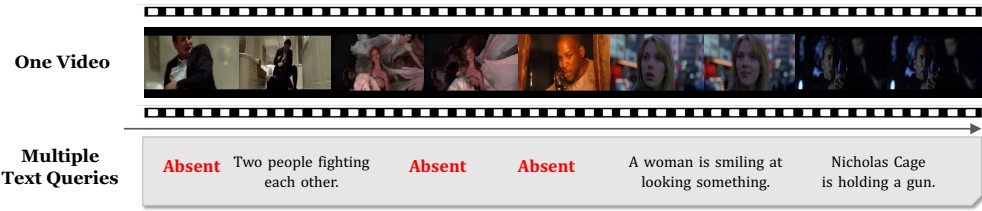

Figure 1: Videos contain much richer information than text. A video can be described by numerous possible text queries, while some of them are missing in the data.

data asymmetry. In contrast, we propose a novel, data-centric approach that directly addresses this issue by enhancing the textual representations at both the training and retrieval stages.

In this paper, we introduce a unified text enrichment framework designed to bridge the information asymmetry between text and video. Our method operates on multiple levels to comprehensively enrich textual representations. During training, we generate event-level captions for videos using pre-trained Vision-Language Models (VLMs) and an event-aware segmentation mechanism, ensuring that all semantically significant moments in the video are captured. This process results in a richer, more diverse set of textual representations that more accurately reflect the video's content. During retrieval, we harness the generative power of Large Language Models (LLMs) to expand and diversify text queries, allowing for better coverage of the semantic complexity of the target video. This two-stage approach ensures that the textual modality can more fully represent the richness of the video, significantly narrowing the information gap.

However, simply generating multiple enriched queries introduces its own challenge: not all queries contribute equally to retrieval success. This realization led us to explore how effectively selecting the most relevant queries can further enhance retrieval performance. A key insight from our work is the introduction of the "Oracle Query" concept, which sheds light on the potential of query selection to significantly boost retrieval performance. By selecting the optimal query from a set of enriched queries — the so-called "Oracle Query" — we observed a dramatic improvement in retrieval accuracy. For example, in the MSR-VTT dataset, the Rank-1 metric increased from 46.7% to 61.4%. While this "oracle" approach requires access to ground truth during query selection (which is impractical in real applications), it reveals a critical opportunity: effectively selecting queries can simultaneously reduce computational cost and enhance retrieval accuracy. To capitalize on this finding, we introduce a novel query selection mechanism, which selects the most relevant and diverse queries without requiring oracle-level information. This mechanism evaluates queries based on their semantic relevance and diversity, ensuring that only the most effective queries are used. By doing so, we achieve a further performance improvement with the reduction in computational cost, further narrowing the gap between text and video representations. Our contributions can be summarized as follows:

- We address the problem of information asymmetry in text-video retrieval from a novel data-centric perspective, enriching text representations to better align with video content.

- We propose a unified text enrichment framework that operates during both the training and retrieval stages, leveraging VLMs for event-level captioning and LLMs for query diversification.

- We introduce the concept of "Oracle Query", demonstrating the performance potential of optimal query selection in narrowing the gap between text and video representations.

- We design a query selection mechanism that optimizes the use of enriched queries, reducing computational costs and improving retrieval accuracy.

- Our method achieves state-of-the-art performance on standard benchmarks such as MSR-VTT (Xu et al., 2016b), MSVD (Wu et al., 2017b), LSMDC (Rohrbach et al., 2015b), and VATEX (Wang et al., 2019), demonstrating its robustness and effectiveness across diverse datasets.

## 2 RELATED WORK

**Text-Video Retrieval** Early text-video retrieval methods primarily relied on pre-trained unimodal models, leveraging hand-crafted features and modeling cross-modal alignment (Gabeur et al., 2020; Wang et al., 2021a;b; Li et al., 2022) The introduction of vision-language models (VLMs), represented by CLIP Radford et al. (2021), have inspired works such as CLIP4Clip (Luo et al., 2022) and Straight-

CLIP (Portillo-Quintero et al., 2021) to build retrieval models by leveraging pre-trained shared latent spaces. Subsequent studies have tried to finetune CLIP on video-text datasets with specialized designs for cross-modal attention mechanisms (Gorti et al., 2022; Wang et al., 2022; Fang et al., 2021; Ibrahimi et al., 2023; Li et al., 2024). VLMs have also inspired works utilizing synthetic data for this task. Cap4video (Wu et al., 2023b) and CLIP-VIP (Xue et al., 2022) focus on novel architectures using global video captions or fine-grained frame captions, while InternVid (Wang et al., 2023b) and HAVTR (Wang et al., 2024b) generate more larger datasets to improve performance. Our approach differs by focusing on data-centric solutions to address information asymmetry problem in text-video retrieval. By comprehensively considering both training and retrieval phase, our approach demonstrates significant improvements without the need for architectural changes or large-scale data augmentation. In addressing information asymmetry, T-MASS (Wang et al., 2024a) model the text embedding in the latent space as a stochastic embedding, enriching it with a flexible semantic range. UAVTR (Fang et al., 2023) proposes a uncertainty-adaptive approach that models each look-up as a distribution matching procedure. However, they do not fully resolve the asymmetry issue in terms of data. Our work takes a novel data-centric perspective to tackle this problem, providing a complementary approach to these models.

**Foundation Models**  The increasing data and compute give birth to large foundation models (FMs) (Bommasani et al., 2021), including unimodal FMs and multimodal FMs. The most representative unimodal FMs is Large Language Models (LLMs) (Touvron et al., 2023; Brown et al., 2020). By training on a huge amount of text data via next-token prediction, LLMs demonstrate exceptional capabilities in language understanding and generation, instruction following, and emergent ability. The development of LLMs has also boosted the vision-language domain in various ways. On the model side, a series of Multimodal Large Language Models (MLLMs) (Wu et al., 2023a; Bai et al., 2024), such as BLIP-2 (Li et al., 2023a), LLaVA (Liu et al., 2023), have been built based on LLMs, demonstrating remarkable performance on vision-language tasks (Wang et al., 2020; Bai et al., 2021) and video tasks (Tang et al., 2023; Bai et al., 2025; Lin et al., 2024; Fan et al., 2023). On the data side, there are some attempts on "re-writing" (Fan et al., 2024) training text data of CLIP (Radford et al., 2021) using LLMs to upgrade the original CLIP model. In contrast to previous methods that focus solely on text generation, our text enrichment framework utilize both VLMs and LLMs to enrich the textual representations, providing a more diverse and contextually rich set of text-video matches.

**Query Expansion**  Query expansion has long been used in document retrieval to bridge the gap between sparse queries and rich document content (Formal et al., 2021). Traditional approaches use lexical knowledge bases (Voorhees, 1994) or pseudo-relevance feedback (Lavrenko & Croft, 2017), while modern methods leverage deep generative models (Zheng et al., 2020). Some recent studies explore prompting LLMs for query expansion (Wang et al., 2023a), aiming to diversify the query to mitigate vocabulary mismatch. Our method diverges from traditional query expansion in a key way: we tackle the more challenging cross-modal scenario by using LLMs not just to mitigate lexical gaps, but to enrich the query's semantic depth and diversity, ensuring the generated queries better align with the rich content of video data. This approach allows us to cover a broader range of possible video descriptions while maintaining high retrieval precision.

## 3 METHOD

### 3.1 MODEL

In text-video retrieval, the goal is to learn a similarity function $s(Q, V)$ for a given text query $Q$ and a video $V$. It aims to maximize the similarity score of positive text-video samples and assign lower similarity for irrelevant pairs. Consistent with previous studies Luo et al. (2022); Portillo-Quintero et al. (2021); Gorti et al. (2022); Wu et al. (2023b), we adopt pre-trained CLIP Radford et al. (2021) as the backbone. In the process of adapting pre-trained CLIP from image to the video domain, a commonly employed technique is mean-pooling, which aggregates frame embeddings into a global video embedding. However, since videos are much more expressive than texts, typical text-agnostic aggregation methods that encompass the entire video may encode irrelevant information not accounted for in the input text. Hence, we utilize text-conditioned pooling from the previous work, X-Pool Gorti et al. (2022).

To elaborate, when provided with the text query $Q$ and the video $V$, we first use CLIP backbone to obtain the text embedding $\mathbf{e}_t \in \mathbb{R}^D$ and frame embeddings $\mathbf{e}_v \in \mathbb{R}^{F \times D}$, where $D$ is the dimension of the latent space, $F$ is the number of frames. Then we employ a learnable cross-attention module

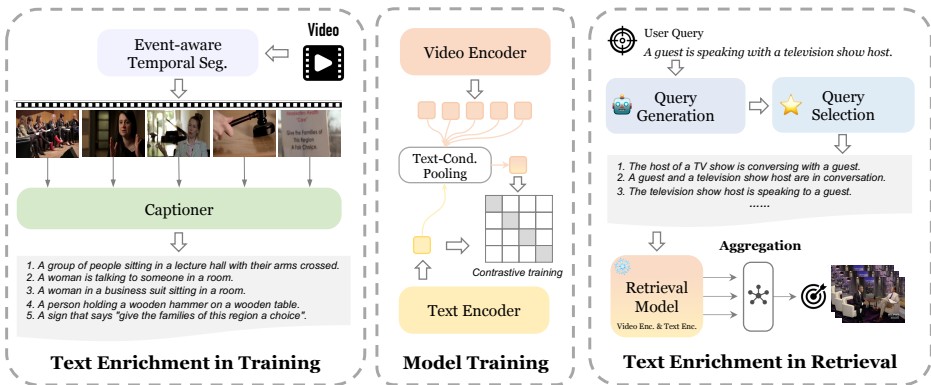

Figure 2: Illustration of the unified text enrichment framework. The left part shows the process of enriching text representations of training data via a comprehensive video captioning approach, including an event-aware video temporal segmentation module and a pre-trained captioner. In the middle, we adopt a dual-encoder model with a text-conditioned video pooling design. The right part illustrates text enrichment during retrieval phase, where a query generation module, a query selection module, and an aggregation module work together to enhance the retrieval performance.

that learns to highlight the most semantically similar video frames as described in the given text:

$$\mathbf{e}_{v|t} = \text{CrossAttn}(\mathbf{e}_t, \mathbf{e}_v, \mathbf{e}_v), \tag{1}$$

where the query of the cross-attention module is the text embedding $\mathbf{e}_t$, the key and value are both frame embeddings $\mathbf{e}_v$. The resulting output $\mathbf{e}_{v|t}$ is an aggregated video embedding conditioned on the text. For the sake of simplicity, we omit certain implementation details, such as layer normalization. For more details, please refer to X-Pool Gorti et al. (2022).

For the training objective, we follow the common practice of optimizing the bidirectional learning objective. A symmetric cross-entropy loss is employed to maximize the similarity between matched text-video pairs and and minimize the similarity for other pairs:

$$\mathcal{L}_{Q2V} = -\frac{1}{B} \sum_{i=1}^{B} \log \frac{\exp(s(\mathbf{e}_t^i, \mathbf{e}_{v|t}^i)/\tau)}{\sum_{j=1}^{B} \exp(s(\mathbf{e}_t^i, \mathbf{e}_{v|t}^j)/\tau))}, \tag{2}$$

$$\mathcal{L}_{V2Q} = -\frac{1}{B} \sum_{i=1}^{B} \log \frac{\exp(s(\mathbf{e}_t^i, \mathbf{e}_{v|t}^i)/\tau)}{\sum_{j=1}^{B} \exp(s(\mathbf{e}_t^j, \mathbf{e}_{v|t}^i)/\tau))}, \tag{3}$$

$$\mathcal{L} = \frac{1}{2}(\mathcal{L}_{Q2V} + \mathcal{L}_{V2Q}), \tag{4}$$

where $s(\cdot, \cdot)$ is a cosine similarity function, $B$ is the batch size and $\tau$ is a learnable scaling parameter.

## 3.2 UNIFIED TEXT ENRICHMENT

### 3.2.1 TRAINING PHASE

The text-conditioned pooling architecture effectively addresses the information asymmetry between the text query and the video by capturing video frames relevant to the text. As a result, irrelevant video information not mentioned in the text will be eliminated. However, asymmetry in the dataset still limits the model performance. For instance, information considered irrelevant for text query A might be highly relevant to text query B. If query B is not present in the training dataset, this information is unlikely to be captured during training. In such a scenario, the video embedding will be biased toward the content described by query A in the latent space. During retrieval, query B will struggle to effectively retrieve the video. If the text queries could comprehensively capture all scenes of a video, then the information of the video can be sufficiently and evenly mined. To tackle this problem, we propose enriching the text queries in the training set using a comprehensive video captioning approach with two key modules: a video temporal segmentation module and an image captioning module.

As the core of the video temporal segmentation module, Kernel Temporal Segmentation (KTS) Potapov et al. (2014); Lin & Lei (2023) algorithm is adopted. KTS divides the video into non-overlapping temporal segments by analyzing a sequence of frame descriptors and identifying the change points. In our implementation, we use a pre-trained CLIP to extract the frame features. Given a video $V \in \mathbb{R}^{F \times H \times W \times 3}$,

$$\mathbf{e}_v = \text{CLIP}_{img}(V), \quad \mathbf{e}_v \in \mathbb{R}^{F \times D}, \tag{5}$$

$$\{t_1, ..., t_{m^*}\} = \text{KTS}(\mathbf{e}_v), \tag{6}$$

where $t_i$ is the change-point position, $m^*$ is the optimal number of change points selected by KTS. For the sake of simplicity, we omit the details of the KTS algorithm. With the detected change-points, the video can be split into multiple segments. Since CLIP features contain semantically rich information, the obtained segments are semantically-consistent, being regarded as 'atomic' scenes.

Next, we employ an image captioning module to generate descriptions for each scene. Specifically, we select the middle frame of each scene as the representative frame as Eq. 7. Compared with video captioning, an image captioning model is more efficient as it takes only one frame as input. We assume a single frame is sufficient for representing the 'atomic' scene.

$$Q_i^{train} = \text{ImageCaptioner}(V[\frac{t_i + t_{i+1}}{2}]). \tag{7}$$

With the above modules, we can generate a set of text queries that captures each semantically-consistent 'atomic' scene for a given video. By running on all the videos of the training set, a video-text dataset with complete text queries is obtained. Inevitably, even using state-of-the-art image captioning models, there will be some noise being introduced. In order to make our approach compatible with a wider range of model-centric methods and reveal the importance of data, we have not explored model architecture designs specifically tailored to address the noise issue. Instead, we show that simply introducing a pre-training stage on the noisy expanded dataset is effective. More study on the data will be elaborated in Sec. 4.2.1.

### 3.2.2 RETRIEVAL PHASE

In traditional text-video retrieval, one text query is used to retrieve the target video. However, under the information asymmetry condition, a video can be successfully retrieved by multiple possible text queries. In this section, we introduce text query enrichment in the retrieval phase.

In this phase, there is no access to the video content or other potential texts related to the video. We must initiate the process from the single existing text query. Large Language Models (LLMs) are powerful text generation models, especially suited for zero-shot, few-shot, or prompt-based learning. Therefore, we design prompts to instruct an LLM to understand the given text description, reason the visual scene, and describe the scene in a diverse way to cover various semantic concepts. The detailed prompts will be provided in the Appendix. Ablations (see Tab. 6) suggest that LLMs are preferable over manual or rule based methods (*e.g.* NLTK Bird et al. (2009)) thanks to its language modeling and reasoning capability. Formally, given a text query $Q$,

$$\{Q_1^{test}, ..., Q_n^{test}\} = \text{LLM}(\text{prompt}, Q), \tag{8}$$

$n$ is the number of generated queries, specified in the prompts. In this paper, we set $n$ to 10.

Next, we explain how the generated queries aid in the retrieval process. One straightforward approach is to create a new, longer text description by appending additional information to the original text. The advantage is that it does not bring additional computation since it only requires one-time similarity computation. However, in our experiments (see Tab. 6), we discovered that concatenating text queries or average-pooling the text embeddings had a detrimental effect on retrieval performance. We hypothesize that the text encoder struggles to understand the long description. Another option is to run the retrieval with each query independently and then aggregate the results (similarity or ranking). By experimenting a series of aggregation strategies, our empirical study shows that aggregating the ranking results using a majority voting strategy yielded the best performance.

### 3.3 QUERY SELECTION MECHANISM

Despite the performance improvement, several drawbacks remain. First, not all enriched text queries contribute equally to retrieval success. Some contain noisy information due to LLM hallucination, while others are nearly identical to the original query, bringing no novel information. Additionally, more queries increase computational costs. For example, using 1 original and 10 enriched queries

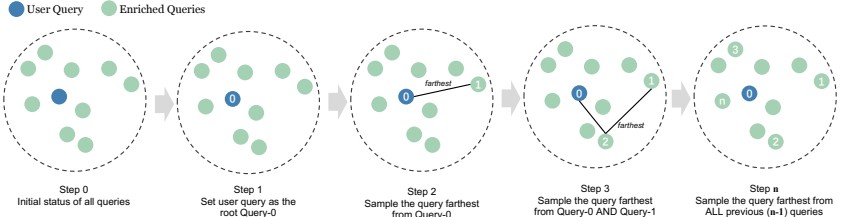

Figure 3: Illustration of Farthest Query Sampling (FQS) algorithm. The queries are distributed within a certain range of relevance. The blue point (user query) is set as the root query. At each step, FQS samples the query that is farthest from *all* previous sampled queries.

requires 11 times more pairwise similarity computations than the previous approach. These challenges motivate us to explore effective query selection.

We first introduce the concept of "**Oracle Query**". From the set of $\{Q, Q_1^{test}, ..., Q_n^{test}\}$, we always select the one that yields the best retrieval performance. Since this process requires access to the ground truth video-text matching, which is impractical in application, we only use this approach to demonstrate the potential of text query enrichment and query selection. It can be regarded as an *upper bound* performance. We found that this oracle query selection can achieve dramatic performance improvement even on a baseline model. In MSR-VTT dataset, by simply replacing the original query with oracle query, the Rank-1 metric is boosted from $46.7\%$ to $61.4\%$ (see Tab. 6). This surprising result reveals two key insight: i) text enrichment in text phase has significant potential for improving text-video retrieval; ii) selecting an effective query can simultaneously reduce computational costs and enhance retrieval performance.

The result motivates us to investigate *effective* query selection. When defining *effective* query selection, our primary considerations are two aspects: *relevance* and *diversity*. The former aspect demands that enriched queries should accurately convey semantic meaning with minimal illusions. The latter emphasizes that the text queries should be expressed in a variety of ways, such that more novel views can be included. Often, these two aspects exhibit some degree of contradiction. Relevance may limit diversity, whereas increased diversity also increases the risk of introducing illusions. In this work, the relevance is controlled in two folds. First, by designing an appropriate prompt, we enforce the LLM to generate enriched text queries strictly adheres to the factual information in the original query, limiting the generated ones in a certain range. Secondly, in the following query selection process, we always primarily include the user query as the initial one, serving as the most relevant and reliable anchor. As shown in Fig. 3, the output queries are constrained within a range of relevance. After that, we sample a certain number of queries from this pool. The goal of this step is to maximize the diversity within this range. In the process, we design a novel Farthest Query Sampling (FQS) algorithm inspired by point sampling in point cloud research Qi et al. (2017).

The main idea is to iteratively select queries in a way that maximizes the minimum distance between the selected queries. The algorithm starts with an initial query and repeatedly selects the query that is farthest from **all** the previously selected queries. This process continues until the desired number of queries is selected. Note that the result of this algorithm is affected by the initial query. We set the original user query as the initial query provided by the human user to stabilize the algorithm and provide double insurance for *relevance*. The formal expression is Eq. 9, where $Q$ is the original query, $\{Q_1^*, ..., Q_k^*\}$ are the enriched queries, $k$ is the number of query selection ($k \leq n$). The distance is computed at an embedding space.

$$\{Q, Q_1^*, ..., Q_k^*\} = \text{FQS}(\{Q, Q_1^{test}, ..., Q_n^{test}\}, k), \tag{9}$$

## 4  EXPERIMENTS

**Setups.** We employ MSR-VTT Xu et al. (2016a), MSVD Wu et al. (2017a), LSMDC Rohrbach et al. (2015a), and VATEX Wang et al. (2019) to evaluate our method. We use standard retrieval metrics: recall at rank K (R@K, higher is better), median rank (MdR, lower is better), and mean rank (MnR, lower is better) to evaluate the performance. The model backbone is initialized from pre-trained CLIP ViT-B/32 and undergo end-to-end training on each dataset. More details of experiment setup and implementation are elaborated in the appendix.

Table 1: Text-to-video retrieval results on the 1K-A test set of MSR-VTT 1K Xu et al. (2016b). *Denotes that the method uses DSL Cheng et al. (2021) post-processing.

| Method | Venue | Rank@1 | Rank@5 | Rank@10 | Median Rank | Mean Rank |
|---|---|---|---|---|---|---|
| *CLIP-ViT-B/32* | | | | | | |
| CLIP4Clip Luo et al. (2022) | Neurocomp.'22 | 44.5 | 71.4 | 81.6 | 2.0 | 15.3 |
| X-Pool Gorti et al. (2022) | CVPR'22 | 46.9 | 72.8 | 82.2 | 2.0 | 14.3 |
| QB-Norm Bogolin et al. (2022) | CVPR'22 | 47.2 | 73.0 | 83.0 | 2.0 | - |
| TS2-Net Liu et al. (2022) | ECCV'22 | 47.0 | 74.5 | 83.8 | 2.0 | 13.0 |
| DRL Wang et al. (2022) | arXiv'22 | 47.4 | 74.6 | 83.8 | 2.0 | - |
| UATVR Fang et al. (2023) | ICCV'23 | 47.5 | 73.9 | 83.5 | 2.0 | 12.9 |
| Cap4Video Wu et al. (2023b) | CVPR'23 | 49.3 | 74.3 | 83.8 | 2.0 | 12.0 |
| TEFAL Ibrahimi et al. (2023) | ICCV'23 | 49.4 | 75.9 | 83.9 | 2.0 | 12.0 |
| ProST Li et al. (2023c) | ICCV'23 | 48.2 | 74.6 | 83.4 | 2.0 | 12.4 |
| UCOFIA Wang et al. (2023c) | ICCV'23 | 49.4 | 72.1 | - | - | 12.9 |
| UMT Li et al. (2023b) | ICCV'23 | 51.0 | 76.5 | 84.2 | - | - |
| CLIP-VIP Xue et al. (2022) | ICLR'23 | 50.1 | 74.8 | 84.6 | 1.0 | - |
| T-MASS Wang et al. (2024a) | CVPR'24 | 50.2 | 75.3 | 85.1 | 1.0 | 11.9 |
| TeachCLIP Tian et al. (2024) | CVPR'24 | 46.8 | 74.3 | - | - | - |
| KDProR Zhuang et al. (2024) | ECCV'24 | 49.6 | 75.1 | 84.4 | 2.0 | 11.6 |
| DITS Wang et al. | NeurIPS'24 | 51.9 | 75.7 | 84.6 | 1.0 | 11.6 |
| **X-Pool Gorti et al. (2022) + Ours** | | 52.1 | 76.8 | 86.3 | 1.0 | 9.7 |
| **X-Pool Gorti et al. (2022) + Ours*** | | 54.1 | 81.6 | **89.7** | 1.0 | **8.0** |
| **CLIP-VIP Xue et al. (2022) + Ours** | | 53.2 | 79.3 | 88.2 | 1.0 | 10.2 |
| **CLIP-VIP Xue et al. (2022) + Ours*** | | **58.9** | **83.4** | 89.5 | **1.0** | 8.3 |
| *CLIP-ViT-B/16* | | | | | | |
| CLIP2TV Gao et al. (2021) | arXiv'21 | 48.3 | 74.6 | 82.8 | 2.0 | 14.9 |
| CenterCLIP Zhao et al. (2022) | SIGIR'22 | 48.4 | 73.8 | 82.0 | 2.0 | 13.8 |
| TS2-Net Liu et al. (2022) | ECCV'22 | 49.4 | 75.6 | 85.3 | 2.0 | 13.5 |
| DRL Wang et al. (2022) | arXiv'22 | 50.2 | 76.5 | 84.7 | 1.0 | - |
| UATVR Fang et al. (2023) | ICCV'23 | 50.8 | 76.3 | 85.5 | 1.0 | 12.4 |
| Cap4Video Wu et al. (2023b) | CVPR'23 | 51.4 | 75.7 | 83.9 | 1.0 | 12.4 |
| CLIP-VIP Xue et al. (2022) | ICLR'23 | 54.2 | 77.2 | 84.8 | - | - |
| T-MASS Wang et al. (2024a) | CVPR'24 | 52.7 | 77.1 | 85.6 | 1.0 | 10.5 |
| **X-Pool Gorti et al. (2022) + Ours** | | 54.6 | 79.5 | 86.7 | 1.0 | 9.1 |
| **X-Pool Gorti et al. (2022) + Ours*** | | **57.1** | **81.9** | **89.2** | **1.0** | **8.0** |

Table 2: Comparisons with SOTA of *t2v* retrieval on MSVD Wu et al. (2017b). Grey text denotes using ViT-B/16 backbone.

| Method | R@1↑ | R@5↑ | R@10↑ | MdR↓ |
|---|---|---|---|---|
| CE Liu et al. (2019) | 19.8 | 49.0 | 63.8 | 6.0 |
| SUPPORT Patrick et al. (2021) | 28.4 | 60.0 | 72.9 | 4.0 |
| CLIP Radford et al. (2021) | 37.0 | 64.1 | 73.8 | 3.0 |
| Frozen Bain et al. (2021) | 33.7 | 64.7 | 76.3 | 3.0 |
| TMVM Lin et al. (2022) | 36.7 | 67.4 | 81.3 | 2.5 |
| CLIP4Clip Luo et al. (2022) | 45.2 | 75.5 | 84.3 | 2.0 |
| UATVR Fang et al. (2023) | 46.0 | 76.3 | 85.1 | 2.0 |
| X-Pool Gorti et al. (2022) | 47.2 | 77.4 | 86.0 | 2.0 |
| DRL Wang et al. (2022) | 48.3 | 79.1 | 87.3 | 2.0 |
| UCOFIA Wang et al. (2023c) | 47.4 | 77.6 | - | - |
| TeachCLIP Tian et al. (2024) | 47.4 | 77.3 | - | - |
| Cap4Video Wu et al. (2023b) | 51.8 | 80.8 | 88.3 | 1.0 |
| **X-Pool Gorti et al. (2022) + Ours** | **51.1** | **81.1** | **88.6** | **1.0** |

Table 3: Comparisons with SOTA of *t2v* retrieval on VATEX Wang et al. (2019). Grey text denotes using ViT-B/16 backbone.

| Method | R@1↑ | R@5↑ | R@10↑ | MdR↓ |
|---|---|---|---|---|
| HGR Chen et al. (2020) | 35.1 | 73.5 | 83.5 | 2.0 |
| CLIP Radford et al. (2021) | 39.7 | 72.3 | 82.2 | 2.0 |
| SUPPORT Patrick et al. (2021) | 44.9 | 82.1 | 89.7 | 1.0 |
| CLIP4Clip Luo et al. (2022) | 55.9 | 89.2 | 95.0 | 1.0 |
| QB-Norm Bogolin et al. (2022) | 58.8 | 88.3 | 93.8 | 1.0 |
| TS2-Net Liu et al. (2022) | 59.1 | 90.0 | 95.2 | 1.0 |
| TEFAL Ibrahimi et al. (2023) | 61.0 | 90.4 | 95.3 | 1.0 |
| ProST Li et al. (2023c) | 60.6 | 90.5 | 95.4 | 1.0 |
| UATVR Fang et al. (2023) | 61.3 | 91.0 | 95.6 | 1.0 |
| UCOFIA Wang et al. (2023c) | 61.1 | 90.5 | - | - |
| T-MASS Wang et al. (2024a) | 63.0 | 92.3 | 96.4 | 1.0 |
| Cap4Video Wu et al. (2023b) | 66.6 | 93.1 | 97.0 | 1.0 |
| **X-Pool Gorti et al. (2022) + Ours** | **63.3** | **92.4** | **96.8** | **1.0** |

## 4.1 COMPARISON WITH THE STATE-OF-THE-ART

In this section, we compare our method with recent state-of-the-art methods on four benchamarks. Tab. 1 shows the comparisons on MSR-VTT. Our method sets a new state-of-the-art performance in text-to-video retrieval for both ViT-B/32 and ViT-B/16 backbones, surpassing previous methods significantly. For instance, we achieve a remarkable +7.6% higher R@1 compared to CLIP4Clip when using the same ViT-B/32 backbone for text-to-video retrieval, without any post-processing techniques. In comparison to methods that also utilize syn-

Table 4: Comparisons with SOTA of *t2v* retrieval on LSMDC Rohrbach et al. (2015b).

| Method | R@1↑ | R@5↑ | R@10↑ | MdR↓ | MnR↓ |
|---|---|---|---|---|---|
| CE Liu et al. (2019) | 11.2 | 26.9 | 34.8 | 25.3 | - |
| MMT Gabeur et al. (2020) | 12.9 | 29.9 | 40.1 | 19.3 | 75.0 |
| NoiseE Amrani et al. (2021) | 6.4 | 19.8 | 28.4 | 39.0 | - |
| Straight-CLIP Portillo-Quintero et al. (2021) | 11.3 | 22.7 | 29.2 | 56.5 | - |
| MDMMT Dzabraev et al. (2021) | 18.8 | 38.5 | 47.9 | 12.3 | 58.0 |
| Frozen Bain et al. (2021) | 15.0 | 30.8 | 39.8 | 20.0 | - |
| TeachText-CE+ Croitoru et al. (2021) | 17.2 | 36.5 | 46.3 | 13.7 | - |
| CLIP4Clip Luo et al. (2022) | 22.6 | 41.0 | 49.1 | 11.0 | 61.0 |
| XPool Gorti et al. (2022) | 25.2 | 43.7 | 53.5 | 8.0 | 53.2 |
| CLIP-VIP Xue et al. (2022) | 25.6 | 45.3 | 54.4 | 8.0 | - |
| ProST Li et al. (2023c) | 24.1 | 42.5 | 51.6 | 8.0 | - |
| TEFAL Ibrahimi et al. (2023) | 26.8 | 46.1 | 56.5 | 7.0 | 44.4 |
| **X-Pool Gorti et al. (2022) + Ours** | **28.3** | **49.2** | **59.1** | **6.0** | **37.1** |

thetic text captions, such as Cap4Video Wu et al. (2023b) and CLIP-VIP Xue et al. (2022), our approach consistently achieves higher performance with both ViT-B/32 and ViT-B/16 backbones. It's worth noting that CLIP-VIP uses the large-scale video-text dataset HowTo100M Miech et al. (2019) for pre-training. Despite this, our method still outperforms it, providing further evidence of effectiveness of our approach. We further show that, as a data-centric method, our method is compatible

Table 5: Ablation study on text enrichment of training phase on MSR-VTT Xu et al. (2016b). We compare our method with other captioning-based text enrichment strategies.

| | Text Enrichment | R@1↑ | R@5↑ | R@10↑ | MdR↓ | MnR↓ |
|---|---|---|---|---|---|---|
| 1 | ✗ (Baseline) | 46.7 | 73.0 | 83.3 | 2.0 | 14.2 |
| 2 | Global Video Caption | 46.9 | 73.0 | 81.9 | 2.0 | 13.9 |
| 3 | Fixed Time Interval Caption | 47.0 | 72.2 | 82.3 | 2.0 | 14.2 |
| 4 | Ours (Training Enrich. Only) | 47.7 | 71.9 | 82.1 | 2.0 | 13.9 |
| 5 | Ours (Retrieval Enrich. Only) | 50.2 | 78.0 | 86.9 | 1.0 | 10.1 |
| 6 | Ours (Unified Enrichment) | 52.1 | 76.8 | 86.3 | 1.0 | 9.7 |

Table 6: Ablation study on text enrichment of the retrieval phase, including query generation and result aggregation strategies.

| Query Generation | Result Aggregation | R@1↑ | R@5↑ | R@10↑ | MdR↓ | MnR↓ |
|---|---|---|---|---|---|---|
| ✗ | ✗ | 46.7 | 73.0 | 83.3 | 2.0 | 14.2 |
| NLTK | Concat. | 40.2 | 67.7 | 78.3 | 2.0 | 17.5 |
| | Avg. Text | 45.1 | 71.8 | 81.6 | 2.0 | 14.7 |
| | Avg. Sim. | 45.4 | 71.7 | 81.6 | 2.0 | 14.9 |
| | Voting | **48.3** | 73.4 | 84.3 | 2.0 | 10.2 |
| | Oracle | 59.3 | 82.1 | 90.5 | 1.0 | 7.2 |
| LLM | Concat. | 45.4 | 70.5 | 80.3 | 2.0 | 15.7 |
| | Avg. Text | 45.9 | 73.6 | 82.3 | 2.0 | 13.7 |
| | Avg. Sim. | 46.1 | 73.7 | 82.2 | 2.0 | 13.7 |
| | Voting | **49.6** | 76.2 | 86.0 | 2.0 | 9.2 |
| | Oracle | 61.4 | 82.4 | 90.0 | 1.0 | 7.4 |

with stronger models, such as CLIP-VIP Xue et al. (2022), and post-processing techniques, such as DSL Cheng et al. (2021), yielding significantly higher performance.

Furthermore, we conduct evaluations on other benchmarks, including MSVD (Tab. 2), VATEX (Tab. 3), and LSMDC (Tab. 4). To ensure a fair comparison, all experiments in these three datasets are conducted using the ViT-B/32 backbone. Our method consistently improves performance across different datasets, ultimately achieving new state-of-the-art results. Overall, our method's consistent performance improvements across these four benchmarks demonstrate the its effectiveness.

## 4.2 ABLATION STUDY

### 4.2.1 TEXT ENRICHMENT OF TRAINING PHASE

In Tab. 5, we study the impact of text enrichment of training phase. Row 1 is the baseline model trained on the original MSR-VTT Xu et al. (2016b), without query enrichment in training. Row 2 shows results with the model pre-trained on video captions provided by a previous work Wu et al. (2023b), which uses a zero-shot video captioning model ZeroCap Tewel et al. (2022). In Row 3, we design a strawman setting, in which we uniformly break a video in fixed intervals and do captioning for each chunk. The text-video retrieval model is then pre-trained on this enriched caption set. In Rows 4 and Row 6, the model is pre-trained on comprehensive, event-aware video captions generated by our approach. To ensure that the scale of data does not lead to unfair comparison, we generate a comparable number of captions for Rows 2 to 6, resulting in approximately 45K additional queries for the 9K training videos in the MSR-VTT dataset. Comparing Rows 1 to 4 suggests that incorporating extra text queries in the training data is generally beneficial. Particularly, Our (Training Enrich. Only) is shown to be more effective, demonstrating a 1.0% performance increase. We attribute this improvement to the proposed approach, which effectively reduces the information gap by covering more video scenes. To further validate effectiveness, we compare Row 5 Ours (Testing Enrich. Only) and Row 6 Ours (Unified Enrichment). The 1.9% performance gap demonstrate the text enrichment of training again.

### 4.2.2 TEXT ENRICHMENT OF RETRIEVAL PHASE

We use 1 original query and 10 enriched queries on the MSR-VTT test set to evaluate and compare different types of query generation and result aggregation methods.

**Query Generation.** We first compare the two types of query generation methods. For the NLTK-based method, we utilize the NLTK Bird et al. (2009) tool to identify representative terms (nouns

Table 7: Ablation study on query selection, including the selection algorithm and the choice of parameter $k$, *i.e.*, number of enriched text queries. We report only R@1 metric for simplicity. The **bold** number denotes the highest number for each **row**.

| Model | Query Selection | $k = 0$ | $k = 2$ | $k = 6$ | $k = 10$ |
|---|---|---|---|---|---|
| Baseline | Random | 46.7 | **49.6** | 48.8 | **49.6** |
| | | 46.7 | **50.2** | 48.9 | 49.6 |
| | | 46.7 | 49.0 | **49.6** | **49.6** |
| | | 46.7 | **49.9** | 49.0 | 49.6 |
| | $k$-DPP Chen et al. (2018) | 46.7 | **50.2** | 49.9 | 49.6 |
| | FQS | 46.7 | **50.2** | 50.1 | 49.6 |
| *w.* Unified Text Enrichment | $k$-DPP Chen et al. (2018) | 46.7 | 50.9 | **51.0** | 49.6 |
| | FQS | 47.7 | **52.1** | 50.5 | 49.6 |

and verbs) in the query and generate new queries by replacing these terms with their synonyms obtained from the NLTK tool. This strategy can be regarded as traditional query expansion. For the LLM-based method, we instruct the LLM (GPT-4) to generate text descriptions by reasoning the visual scene. As shown in Tab. 6, under all the result aggregation methods, LLM always outperforms the NLTK-based method. Notably, the oracle result, which is achieve by using **one single text query** and removes the influence of aggregation methods, provide a clearer distinction between the two query generation methods. We will elaborate the detail of how we achieve the oracle performance in Appendix. This performance gap demonstrates the effectiveness of using LLM as a tool for text query enrichment, thanks to its exceptional capabilities on high-level language understanding, reasoning, and generation.

**Result Aggregation.** Next, we investigate different methods for aggregating the retrieval results from multiple queries. The results in Tab. 6 show that certain methods, such as concatenating queries and mean-pooling of text embeddings, can lead to even worse performance than the baseline. Among these methods, majority voting is a simple yet effective one, outperforming other methods. The most interesting part is the Oracle result. Firstly, it shows the huge potential of text enrichment in retrieval phase with the significant performance improvement. Secondly, the gap between the Oracle result and other methods reveals that there are still a large room for improvement. Finally, the Oracle result is achieved without compromising on computational cost, while majority voting with all the 11 queries appears to be inefficient. These findings inspire us to develop a dedicated query selection module to further improve retrieval performance as well as reduce computational cost.

### 4.2.3 IMPACT OF QUERY SELECTION MECHANISM

Query selection aims to enhance retrieval performance to approach the Oracle performance and improve inference efficiency. In Tab. 7, we report the R@1 retrieval performance under various query selection strategies and selection number $k$. Firstly, we employ a random selection strategy on the baseline model and run the retrieval with 4 different random seeds, resulting the first 4 rows of the table. It can be observed that randomly selecting a specific number of queries can lead to promising performance. For example, when $k = 2$, the R@1 accuracy can achieve $49.6$ or even $50.2$. The performance variation among different $k$ indicates that not all the enriched queries contribute equally.

The primary drawback of random query selection is its lack of stability, making the retrieval result of each round different. This characteristic should be avoided in retrieval systems. We demonstrate that the our proposed Farthest Query Sampling achieves competitive or better performance under the same setting of $k$. The performance improvement can be attributed to that our query selection method considers both relevance and diversity of text queries, being able to cover a larger area in the embedding space and naturally increasing successful retrieval rate.

We also compare our FQS with another deterministic method, named determinantal point processes (DPPs) Kulesza et al. (2012). In particular, we utilize MAP inference of DPPs introduced in Chen et al. (2018) for fast and deterministic inference. DPP is similar to our FQS in implementation. It returns a subset of points of a pre-specified size (*i.e.* $k$) that are maximally diverse. The results show that $k$-DPP can be a viable implementation of query selection. However, it exhibits a slightly worse performance than FQS. We analyze that the main difference lies in the initial text query. In FQS, we always take the user query as the starting point and probe enriched queries to maximize the diversity. The involvement of ground-truth user query essentially ensures the relevance of text query and establishes a solid retrieval lower bound, allowing FQS to achieve better retrieval performance. In contrast, $k$-DPP treat the ground-truth user query and the enriched queries equally. It maximizes

Table 8: Experiments conducted on MSR-VTT Xu et al. (2016b) dataset, assessing the impact of LLM employed in retrieval text query enrichment.

| LLM | # Param. | Text Query | R@1 | R@5 | R@10 |
|---|---|---|---|---|---|
| - | - | GT User Query | 47.7 | 71.9 | 82.1 |
| GPT-4 | Unknown | Farthest Enriched Query | 41.6 | 66.2 | 76.3 |
| Phi-3.5 | 3.8B | Farthest Enriched Query | 36.7 | 62.1 | 73.7 |
| GPT-4 | Unknown | FQS ($k$=2) | 52.1 | 76.8 | 86.3 |
| Phi-3.5 | 3.8B | FQS ($k$=2) | 51.5 | 77.7 | 86.5 |

the diversity aspect from a global view, while neglecting the relevance aspect. Thus, it is possible the user query would not be selected. making the retrieval be sensitive to some noisy enriched queries.

In the context of query selection, the hyper-parameter $k$ plays a crucial role. Our results indicate that the optimal value for $k$ is 2. It achieves the highest R@1 score among most of the tested settings as shown in Tab. 7. Notably, when compared with $k = 10$ (using all the queries), setting $k = 2$ results in higher performance, with an R@1 score of 50.2 compared to 49.6, while also substantially reducing computational costs. We choose an even number for $k$ to ensure there is an odd number of queries when including the original query, which avoids potential ties in the majority voting process. In both theoretical and experimental scenarios, $k = 2$ proves to be the optimal choice.

### 4.2.4 IMPACT OF LLM CHOICE IN RETRIEVAL PHASE

In the retrieval phase, the text queries are enriched by an LLM, which plays a crucial role. Throughout the paper, we use GPT-4 API to achieve the text enrichment. In this section, we use an open-source, light-weight LLM, Phi-3.5 Abdin et al. (2024) to do the text enrichment task in retrieval phase. The experiment results are shown in Table 8.

Firstly, when using the farthest one of the enriched queries, we observe that both GPT-4 and Phi-3.5 exhibit inferior retrieval performance compared to using GT user query. The performance gap indicates that naively using LLM to enrich the text query would introduce noise. In the two LLMs, Phi-3.5 shows much worse performance than GPT-4. This aligns well with our expectation, as the Phi-3.5 model is known to be weaker than GPT-4, introducing more noise in the enriched text queries.

Next, we apply our proposed Farthest Query Sample algorithm to the enriched text queries and conduct retrieval. Surprisingly, we observe only a small performance gap between the two LLMs, 0.5% in Rank-1. This experiment proves that our method is robust to the LLM. The FQS algorithm and the majority voting aggregation can mitigate large noise. Moreover, the exceptional performance of Phi-3.5 unveil of opportunity of using small language model, further improving inference efficiency.

### 4.3 MORE RESULTS

We provide more results in the appendix, including A.1: Experiment setups and implementation details; A.3: Details and more interesting findings of the oracle query selection; A.4: Analysis of computational efficiency; A.5 and A.6: experiments on model architectures beyond X-Pool showing that our data-centric approach is compatible with a wide range of model-centric approaches; A.7: prompt in LLM; A.9: qualitative examples that demonstrate how the enriched text queries in retrieval aid in text-video retrieval. We also thoroughly discuss the limitation and future work.

## 5 CONCLUSION

In this work, we address the information asymmetry problem between video and text in the text-video retrieval task. We take a novel data-centric perspective to address this issue at multiple stages. During training, we enrich text descriptions through an event-aware, comprehensive video captioning approach, ensuring full video scene coverage. In the retrieval process, we use large language models to enrich the text queries by reasoning the semantic concepts. We introduce the concept of Oracle Query to reveal the significant potential of data-centric investigation in retrieval. Moreover, we design a farthest query sampling algorithm for effective query selection, improving retrieval performance and reducing computational costs. Extensive experiments prove that our method has consistent improvements over four benchmarks, surpassing current state-of-the-art methods. Our work serves as an inspiration for future research efforts, encompassing both model-centric methods and data-centric enhancements.

ACKNOWLEDGEMENT

This research is supported by the National Research Foundation, Singapore under its AI Singapore Programme (AISG Award No: AISG3-RP-2022-030).

ETHICS STATEMENT

This research does not involve human subjects or the collection of personally identifiable information. The datasets used in this work are publicly available and have been previously vetted for ethical use in academic research. We ensured compliance with relevant data licensing agreements and conducted all experiments in accordance with the terms of use of the datasets. The proposed framework that involves the usage of a large language model (LLM), was designed to improve the fairness and accuracy of cross-modal retrieval systems. However, we acknowledge that LLMs may inadvertently encode and propagate biases present in the training data. To mitigate this, we adopted a data-centric approach aimed at generating diverse and representative queries. We encourage future work to continue monitoring and addressing bias in these models to prevent unintended societal harm. Additionally, our method was designed with computational efficiency in mind to reduce resource consumption and ensure a more environmentally sustainable research practice. No conflicts of interest or external sponsorship influenced the outcomes of this research.

REPRODUCIBILITY STATEMENT

We have made every effort to ensure that our results are fully reproducible. Detailed descriptions of the proposed framework, including the data processing pipeline, model architecture, and training procedures, are provided in the main paper as well as the comprehensive appendix. To facilitate reproducibility, all datasets used in our experiments are publicly datasets, and the specific processing steps are sufficiently described. We will make the code, data, and pre-trained model public.

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

# A  APPENDIX

## A.1  EXPERIMENT SETUPS AND IMPLEMENTATION DETAILS

**Datasets.** MSR-VTT Xu et al. (2016a) contains a total of 10K video clips, each having 20 captions. We utilize the `Training-9k` subset for training and the `test-1K-A` subset for evaluation. MSVD Wu et al. (2017a) contains 1,970 videos with 80K captions, with 40 captions on average per video. There are 1200, 100, and 670 videos in the train, validation, and test sets, respectively. LSMDC Rohrbach et al. (2015a) consists of 118,081 video clips sourced from 202 movies with one caption corresponding to each clip. Evaluation is conducted on a test set of 1,000 videos from movies disjoint from the train and validation sets. VATEX Wang et al. (2019) collects around 35K videos with multiple text annotations in both English and Chinese for each video. There are around 26K videos for training, 1,500 for validation, and 1,500 for testing.

**Evaluation Metrics.** We use standard retrieval metrics: recall at rank K (R@K, higher is better), median rank (MdR, lower is better), and mean rank (MnR, lower is better) to evaluate the performance. R@K (Recall at K) calculates the percentage of test samples for which the correct result is found in the top-K retrieved points to the query. We report results for R@1, R@5, and R@10. Median Rank calculates the median of the ground-truth results in the ranking. Mean Rank calculates the mean rank of all correct results.

**Implementation Details.** We initialize our backbone from the pre-trained CLIP ViT-B/32. Following previous work Gorti et al. (2022), the text-conditioned pooling module is randomly initialized. Our models undergo end-to-end training on each dataset. For the training hyper-parameters, we also follow X-Pool Gorti et al. (2022). The model is trained for 3 epochs on the enriched training set and 5 epochs on the standard training set. A cosine scheduler Loshchilov & Hutter (2016) is employed to decay the learning rate. For all experiments, we follow previous works by uniformly sampling 12 frames from each video and resizing them to 224x224. For text enrichment in training, we use `blip2-opt-2.7b-coco` as the captioner. For text enrichment in retrieval, we utilize GPT-4 model through the API [1] to generate queries. We train our model using NVIDIA A10 24G GPU. The training takes around 12 hours.

## A.2  ADDITIONAL ABLATION OF FQS

Table 9: Comparing FQS with other variants.

| Query Selection (k=2) | Rank-1 | Rank-5 | Rank-10 |
|---|---|---|---|
| Random | 49.6 | 76.1 | 85.9 |
| Nearest Neighbor | 50.4 | 76.0 | 85.7 |
| Farthest Neighbor | 51.4 | 76.5 | 85.8 |
| k-DPP | 50.9 | 76.7 | 85.8 |
| FQS (s=0.5) | 52.1 | 76.8 | 86.3 |
| FQS (s=0.75) | 52.1 | 76.8 | 86.3 |
| FQS (s=0.85) | 51.6 | 76.7 | 86.2 |
| FQS (s=0.95) | 49.8 | 76.1 | 86.3 |
| NQS | 49.9 | 75.8 | 86.1 |
| FQS | 52.1 | 76.8 | 86.3 |

The Farthest Query Sampling (FQS) algorithm proposed in the main paper plays an important role on selecting relevant yet diverse queries. In this section, we compare this algorithm with a wide range of settings to further demonstrate its effectiveness. The results are shown in Tab. 9.

- **Random** serves as a baseline, which randomly select $k$ queries from the enriched queries.

- **Nearest Neighbor** selects $k$ queries that are closest to the user query from the enriched queries.

- **Farthest Neighbor** selects $k$ queries that are farthest to the user query from the enriched queries.

---

[1]`https://openai.com/blog/openai-api`

- $k$-**DPP** is classical algorithm selecting a subset with maximal diversity from a set of data points.
- **FQS(s=)** is an variant of FQS suggested by one reviewer during the review process of the paper. The value of "s" is a minimum similarity threshold. In implementation, when iteratively selecting queries in FQS, we add an additional condition that requires the selected queries to meet the minimum similarity. If no query meets this condition, we will use the original user query instead.
- **NQS** is another variant of FQS suggested by the reviewer, denoting Nearest Query Sampling, which minimizes the minimum distance between selected queries.

The comparison in Tab. 9 can further verify the effectiveness of the proposed FQS algorithm.

### A.3   UNDERSTANDING ORACLE QUERY SELECTION

The oracle query selection is implemented as follows: for one specific instance, among 1 GT query and $k$ enriched queries, we always select the query that yields the best ranking, where the best ranking refers to that the target video is ranked highest. If there are multiple such queries, we prioritize the GT query. We call it 'Oracle' because we need to access the ground-truth matching during the query selection, which is impractical in real application. Following the rule above, we stats the percentage of GT query *vs.* enriched query in the selected oracle query. The statistic is conducted on $1,000$ test samples of MSR-VTT dataset, $k = 10$. The result shows that $59.5\%$ of selected queries come from the GT query set, while $40.5\%$ of selected queries are from the enriched queries. This actually provides an explanation of the huge performance gap between baseline and oracle, suggesting that the text queries can be specially investigated to enhance the retrieval.

### A.4   ANALYSIS OF COMPUTATIONAL COST

Table 10:  Analysis of computational cost.

| Model | Term | $N_t$ | $N_e$ | $N_v$ | Memory | Infer. time |
|---|---|---|---|---|---|---|
| Txt Enc. | $C_{txt\_enc}$ | 1 | 0 | - | 602 MB | 6.30 ms |
| Txt Enc. (Text Enrich.) | $(1 + N_e) \otimes C_{txt\_enc}$ | 1 | 10 | - | 618 MB | 8.65 ms |
| Similarity | $C_{sim}$ | 1 | 0 | 100,000 | 10096 MB | 96.71 ms |
| Similarity (Text Enrich.) | $(1 + N_e) \otimes N_v \otimes C_{sim}$ | 1 | 10 | 100,000 | 16258 MB | 233.80 ms |
| Similarity (Query Select.) | $(1 + N_e) \otimes N_v \otimes C_{sim}$ | 1 | 2 | 100,000 | 11490 MB | 139.27 ms |
| Full Infer. (Baseline) | $C_{total} - C_{LLM}$ | 1 | 0 | 100,000 | - | 103.01 ms |
| Full Infer. (Text Enrich.) | $C_{total} - C_{LLM}$ | 1 | 10 | 100,000 | - | 242.45 ms |
| Full Infe. (Query Select.) | $C_{total} - C_{LLM}$ | 1 | 2 | 100,000 | - | 147.92 ms |

We decompose the computational cost of a CLIP-based retrieval model into several parts. Formally, given $N_t$ text queries and $N_v$ candidate videos, supposing each text query is enriched with $N_e$ additional text queries, the computational cost can be expressed as follows:

$$C_{total} = (N_t + N_t \times N_e) \otimes C_{txt\_enc} + N_v \otimes C_{vid\_enc} + \\ (N_t + N_t \times N_e) \otimes N_v \otimes C_{sim} + N_t \otimes C_{LLM}, \tag{10}$$

where $C_{txt\_enc}$ and $C_{vid\_enc}$ are the inference cost of the text encoder and video encoder, respectively. $C_{sim}$ is the cost of computing the cross-modal similarity. $C_{LLM}$ is the inference cost of LLM. The operation of $\otimes$ indicates that the the cost $C$ is influenced by the factors $N$. Note that $\otimes$ is different from traditional multiplication operation $\times$, as it can be parallelized and accelerated by GPU.

In practical applications, we usually use one query to search among a large volume of candidate videos. So, we first set $N_t = 1$ for simplicity. Besides, the video embeddings are usually pre-extracted and stored in a database. Thus, the $C_{vid\_enc}$ can be omitted. For each query, the total cost can be simplified as:

$$C_{total} = (1 + N_e) \otimes C_{txt\_enc} + (1 + N_e) \otimes N_v \otimes C_{sim} + C_{LLM}. \tag{11}$$

Based on this, we suggest two ways to reduce the computational cost. First, our query selection mechanism can effectively reduce $N_e$ without sacrificing performance. In fact, it even further boost

the performance by a large margin. It is worth mentioning that in real world applications, $N_v$ can be super large. Thus, reducing $N_e$ would be effective on improving the overall efficiency. Second, our approach exhibits strong robustness to the selection of LLM, as discussed in Sec 4.2.4 of the paper. It implies that we are allowed to employ lightweight LLM to reduce $C_{LLM}$ and improve efficiency.

To help understand the inference cost more intuitively, we provide an example in Table 10. In a practical retrieval scenario, we may have one user query and a large volume of candidate videos, say $N_t = 1$ and $N_v = 100,000$. The inference time are computed by averaging 1,000 randomly generated examples. We observe that, with the help of GPU parallelization, increasing $N_e$ does not lead to significant inference burden in text encoder. However, the similarity computation cost a long inference time. This is because X-Pool requires a text-conditioned pooling, as introduced in the main paper. Finally, we show that vanilla text enrichment strategy increase the per query inference time from $103.01ms$ into $242.45ms$. With the help of query selection, the inference time decrease to $147.92ms$, validating the effectiveness of query selection. Not that this inference time does not take $C_{LLM}$ into account, as it varies depends on different LLMs. We acknowledgement that the inference cost of LLM could be large. However, it can be reduced by selecting lightweight model or more advanced techniques, e.g., KV Cache Pope et al. (2023). Our main goal is to unveil the power of data in text-video retrieval. We leave this optimization as a future work.

## A.5 EXPERIMENT ON VANILLA MODEL

In our main paper, we utilize the pre-trained CLIP and the text-conditioned pooling module as the retrieval model. The text-conditioned pooling is designed to capture video frames relevant to the text query, effectively alleviating the information asymmetry. In this section, we demonstrate that our data-centric approach is not coupled with a specific architecture design, such as the text-conditioned pooling. We conduct experiments on the vanilla CLIP model with a mean-pooling strategy to obtain the video embedding. The results are reported in Tab. 11.

Table 11: Text-to-video retrieval results with vanilla CLIP mean-pooling model on MSRVTT Xu et al. (2016b).

| # | Text Enrich. Training | Text Enrich. Retrieval | R@1 | R@5 | R@10 | MdR | MnR |
|---|---|---|---|---|---|---|---|
| 1 | ✗ | ✗ | 42.2 | 70.4 | 79.2 | 2.0 | 15.7 |
| 2 | ✓ | ✗ | 42.9 | 70.2 | 79.8 | 2.0 | 16.3 |
| 3 | ✗ | ✓ | 46.5 | 75.9 | 85.1 | 2.0 | 11.6 |
| 4 | ✓ | ✓ | 48.8 | 75.3 | 84.6 | 2.0 | 11.4 |

The comparison between Row 1 and Row 2, as well as between Row 3 and Row 4, illustrate the effectiveness of Text Enrichment in Training. In the baseline model (Row 1 vs. Row 2), Text Enrichment in Training achieves an improvement of 0.7% on R@1. The performance gain becomes more obvious when combined with Text Enrichment in Retrieval (+2.3% on R@1, Row 3 vs. Row 4). Finally, the 6.6% performance gap between the results in Row 1 and Row 4 further validates the benefits of our unified framework.

## A.6 TEXT ENRICHMENT IN RETRIEVAL AS A TRAINING-FREE TOOL

In this section, we demonstrate that the proposed Text Enrichment in Retrieval can be applied to a wide range of text-video retrieval models as a training-free tool for performance improvement. We conduct evaluations on the MSR-VTT dataset Xu et al. (2016b) using four widely recognized methods. 1) For CLIP Radford et al. (2021) zero-shot, we utilize the pre-trained CLIP model and employ mean-pooling to obtain the video embedding in a zero-shot manner; 2) CLIP4Clip Luo et al. (2022), a classical method in the text-video retrieval domain; 3) X-Pool Gorti et al. (2022), adopted as the base model in our work; 4) CLIP-VIP Xue et al. (2022), recognized as one of the current state-of-the-art methods. To maintain consistency, all experiments utilize the ViT-B/32 as the retrieval backbone and CLIP text encoder for query selection. The baseline results are reproduced from their released codebases.

The text-to-video retrieval results are reported in Tab. 12. We observe that Text Enrichment in Retrieval consistently improves the retrieval performance across the four methods. For the zero-

Table 12: Text-to-video retrieval results with various methods on the MSR-VTT Xu et al. (2016b). The result are reproduced based the released codebase of each method respectively.

| Model | R@1 | R@5 | R@10 | MdR | MnR |
|---|---|---|---|---|---|
| CLIP$_{zero-shot}$ Radford et al. (2021) | 31.2 | 52.4 | 63.6 | 5.0 | 42.8 |
| CLIP$_{zero-shot}$ Radford et al. (2021) + Text Enrich. Retrieval | 36.4 (**+5.2**) | 62.6 | 71.6 | 3.0 | 27.7 |
| CLIP4Clip$_{mean-pooling}$ Luo et al. (2022) (Reproduced) | 42.3 | 70.6 | 81.9 | 2.0 | 16.5 |
| CLIP4Clip$_{mean-pooling}$ Luo et al. (2022) + Text Enrich. Retrieval | 46.3 (**+4.0**) | 76.3 | 85.3 | 2.0 | 11.9 |
| X-Pool Gorti et al. (2022) (Reproduced) | 46.7 | 73.0 | 83.3 | 2.0 | 14.2 |
| X-Pool Gorti et al. (2022) + Text Enrich. Retrieval | 50.6 (**+3.9**) | 76.3 | 86.5 | 1.0 | 10.4 |
| CLIP-VIP Xue et al. (2022) (Reproduced) | 49.3 | 74.8 | 84.9 | 2.0 | 13.4 |
| CLIP-VIP Xue et al. (2022) + Text Enrich. Retrieval | 53.2 (**+3.9**) | 79.3 | 88.2 | 1.0 | 10.2 |
| CLIP-VIP Xue et al. (2022) + DSL Cheng et al. (2021) (Reproduced) | 55.1 | 78.9 | 86.6 | 1.0 | 11.1 |
| CLIP-VIP Xue et al. (2022) + DSL Cheng et al. (2021) + Text Enrich. Retrieval | 58.9 (**+3.8**) | 83.4 | 89.5 | 1.0 | 8.357 |

shot CLIP, our method boosts the R@1 by 5.2% and R@5 by over 10%, which is a significant improvement. Notably, for the state-of-the-art CLIP-VIP Xue et al. (2022), despite the remarkable results achieved, our method still improves the performance by a large margin, further validating its effectiveness.

We note that several popular methods, including DSL Cheng et al. (2021) and QB-Norm Bogolin et al. (2022), enhance pre-trained text-video retrieval performance using a similar training-free approach. In the final experiment of Tab. 12, we demonstrate that our Text Enrichment in Retrieval is compatible with these training-free methods. The combination of our approach and DSL significantly elevates the performance to a much higher level. Moreover, it is important to note that DSL Cheng et al. (2021) requires access to all queries during testing, which is an impractical assumption for real-world retrieval systems. In contrast, our approach does not require access to other queries during testing, making it more practical in real-world applications.

---

**Prompts**

You are given a caption describing a visual scene. Your task is to rewrite the caption into 10 different sentences following the rules:

1. You can diversify the sentence structure and word usage, but you should strictly keep the same semantic meaning.
2. Do not add uncertain details that do not associate with the visual scene. The rewriting should strictly follow the factual information in the original caption.
3. The rewritten captions should be diverse in number of words.
4. The rewritten captions should be no more than 10 words longer than the original caption.

The input caption is: {}

---

Figure 4: Prompts used for text enrichment in retrieval.

## A.7 PROMPT FOR TEXT ENRICHMENT IN RETRIEVAL

We provide the prompt used for enriching testing retrieval queries in Fig. 4.

## A.8 LIMITATION AND FURTHER WORK

Despite the significant performance achieved by our method, it exhibits certain limitations that serve as inspiration for future research. Firstly, in the text enrichment in training, our approach focuses on a data-centric approach that enriches the dataset. However, there is a potential for further improvement by designing dedicated model architectures tailored to the enriched dataset, such as multi-instance learning. Secondly, in the text enrichment of testing, we employ LLM as the tool to generate diverse queries. Given the vast prompting space of LLM, we have not invested significant efforts in prompt engineering. As a future research avenue, more dedicated prompts, such as in-context learning, can be harnessed to further enhance the quality of text queries. Thirdly, although our query selection mechanism has effectively reduced the computational cost in retrieval, the computation of LLM still exists. Optimization on this part can further benefit more practical applications. Fourthly, the

involvement of VLMs and LLMs would inevitably introduce hallucinated content, which may limit the performance of the method. Although the mechanisms in the paper are shown to be resilient to noise, reducing hallucination has the potential to further improve the performance in the future. Finally, we acknowledge that there is still a noticeable gap in terms of retrieval performance and computational cost between our method and the upper bound. This discrepancy encourages further research to bridge the gap.

## A.9 QUALITATIVE EXAMPLES

In this section, we present a series of text-video retrieval examples from the MSR-VTT dataset Xu et al. (2016b). These examples not only qualitatively validate the effectiveness of our approach but also demonstrate how the enriched queries aid in text-video retrieval. The examples from training set are shown in Fig. 9. The examples from the testing phase are shown in Fig. 5, Fig. 6, Fig. 7, and Fig. 8. In each example, we show that the original query fails to retrieve the target video, while the enriched query does. Detailed explanations are presented in the caption of each figure, respectively.

Original query: a soccer team walking out on the field.
✗ Incorrect video.

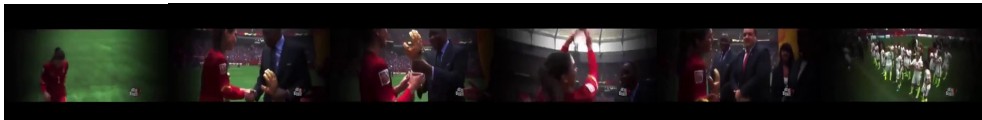

Enriched query: Players from a soccer team are stepping onto the playing field.
✓ Correct video.

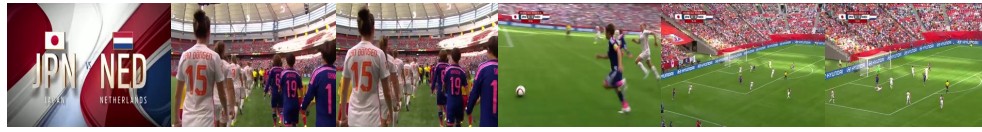

Figure 5: **Rephrasing concepts.** The original query focuses on the concept '*team*'. Although the concept of '*team*' may imply the involvement of '*multiple players*', it is not explicitly stated. The enriched query provides a more explicit view to elaborate the query, correcting the retrieval.

Original query: a woman is cooking food and a man is setting a table.
✗ Incorrect video.

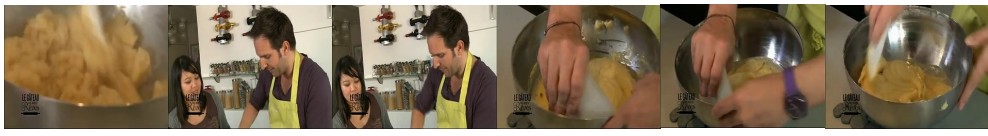

Enriched query: The man is setting a table, and at the same time, a woman is cooking.
✓ Correct video.

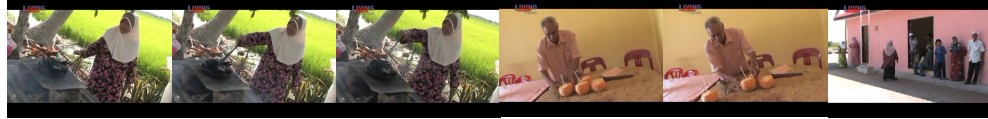

Figure 6: **Altering word order.** The initially recalled video successfully covers the concept of '*woman*', '*man*', and '*cooking food*'. While it does not contain content corresponding to '*setting a table*'. Our hypothesis is that this is because '*setting a table*' is positioned far back in the original query, limiting its impact. The enriched query recalls the video that captures the most terms in the query. This is achieved by altering the word order and emphasis of the original query.

Original query: opening of a nest a **rate** is coming out and searching something it eats something on a human hand

× Incorrect video.

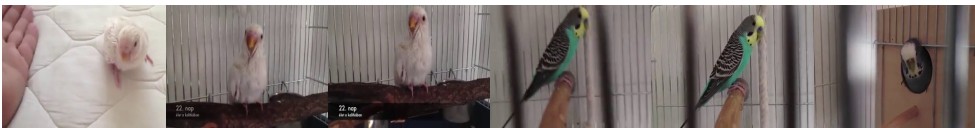

Enriched query: From an open nest, a rat comes out, searches, and consumes something from a human hand.

✓ Correct video.

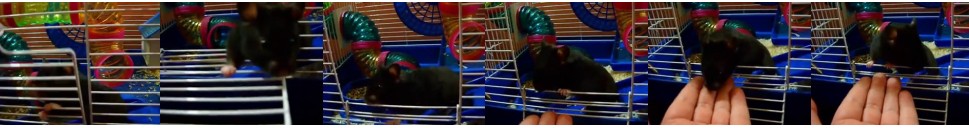

Figure 7: **Typo correction.** We notice that there is a typo in the original query ('*rate*' vs. '*rat*'). Even a very small typo like this can lead to problematic retrieval result. The incorrect video has captured the concept of '*hand*' but fails on other aspects. With the corrected query, we can successfully find the target video. This capability stems from the strong language understanding ability of the large language model, which automatically recognizes the incoherent word in the query and corrects it. This case is challenging to correct for rule-based methods, such a NLTK, since '*rate*' is not a wrong word.

Original query: a woman dances in the background while a guy doesn't move.

× Incorrect video.

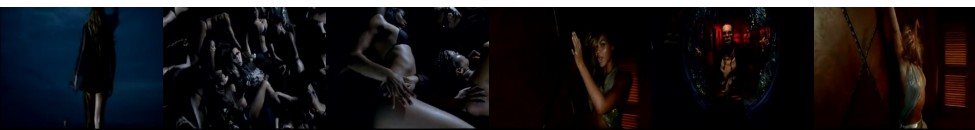

Enriched query: A guy is standing still while a woman in the background is dancing.

✓ Correct video.

Figure 8: **Robustness.** In this example, the original query is clear while the initially recalled video appears to be not strongly relevant to the original query. This might be due to the inherent weakness of the retrieval model. However, the enriched query can still successfully retrieve the target video. This can be seen as an indication that text enrichment improves the robustness of the retrieval model, making it resilient to corner failure cases.

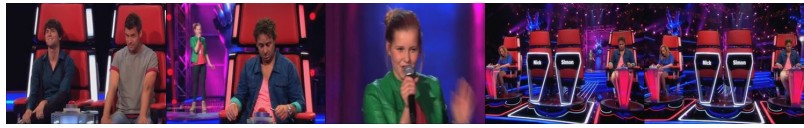

- Two men sitting on red chairs in a room.
- A man sitting in a chair with a microphone and a woman standing behind him.
- A woman in a green jacket holding a microphone.
- The voice judges are sitting in chairs on a stage.

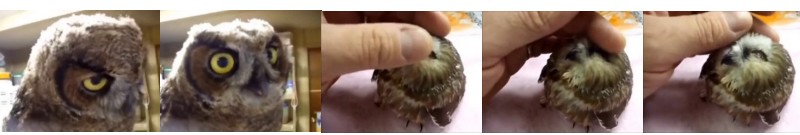

- A close up of an owl with yellow eyes in a room.
- A close up of an owl looking at the camera.
- A person petting a small bird on a pink surface.
- A person is petting a small bird on a pink surface.

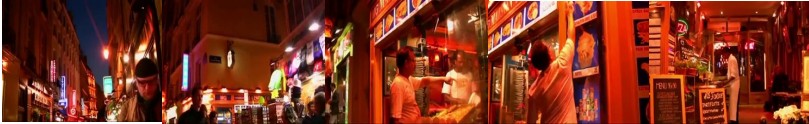

- A man is walking down a city street at night.
- A group of people standing outside of a store at night.
- A man is standing in front of a pizza shop window.
- A man is standing in front of a vending machine.
- A man standing behind a counter in a restaurant with a sign.

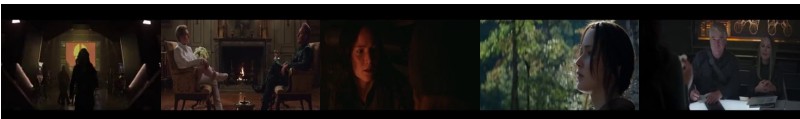

- A person walking down a hallway with a flashlight.
- Two people sitting in chairs talking to each other.
- A woman is looking at something in the dark.
- A woman is looking off into the distance in a forest.
- A man and woman sitting at a table with a laptop.

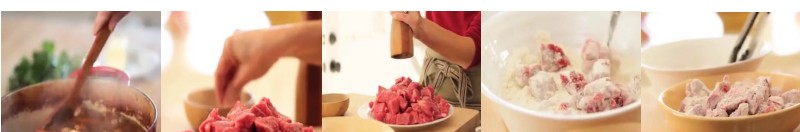

- A person stirring a pot of stew with a wooden spoon.
- A close up of a bowl of meat on a table.
- A person is adding seasoning to a pile of meat.
- A bowl of food with a spoon in it.

Figure 9: Examples from MSR-VTT training dataset with text enrichment.

