# OpenReview forum: "Bridging Information Asymmetry in Text-video Retrieval: A Data-centric Approach"
_ICLR.cc/2025/Conference — ICLR 2025 Poster_

### Official Review · Reviewer_8p6G · 2024-11-03

**Soundness:** 3
**Presentation:** 4
**Contribution:** 4
**Rating:** 8
**Confidence:** 5

**Summary:**

This paper investigates the problem of information asymmetry in text video retrieval from a data-centric perspective. Existing works usually use model-centric approaches of carefully designing advanced text-video interaction modules, e.g., text-conditioned video representations, stochastic embedding, etc. This paper proposes a unified text enrichment framework to enhance the textual representation during both training and testing phase. The results consistently show promising improvements over existing methods. Interestingly, the concept of “oracle query” clearly demonstrate the potential of the query generation and selection, possibly opening up a new venue in this domain.

**Strengths:**

- The paper investigates an under-explored area in the field of text-video retrieval, a data-centric approach to improving the performance by identifying pitfalls in the current training dataset captions and test retrieval process.
- The full method demonstrate remarkable performance in the text-video retrieval task across several benchmarks.
- The experiments with the oracle queries show a huge gap in the current methods, which is novel and interesting. It opens up future works to consider such data-centric approaches for improving performance.
- The paper is well written and easy to follow.

**Weaknesses:**

- There are some existing works on augmenting the data, e.g., [1], what is the main difference between this work and existing works?
- The oracle query experiment shows huge performance gap. Although the result is inspiring, it is better to provide more explanations and/or investigations about this phenomenon. What contribute to the final performance?
- In the ablation study, the performance gain seems mainly come from the Retrieval Phase Enrichment, while the gain Training Phase Enrichment seems to be marginal.
- The method will inevitably increase the computational cost, which should better be investigated.
- It is mentioned that majority voting over the enriched queries yields the best performance. However, it is not clear how this is implemented. I would suggest include such details, as least in the appendix.

[1] HAVTR: Improving Video-Text Retrieval Through Augmentation Using Large Foundation Models. ECCV 2024.

**Questions:**

- Will the author open-source the enriched dataset and code?

---

> ### Author Response · Authors · 2024-11-21
>
> We sincerely thank the reviewer for reviewing our paper. We are glad to see the reviewer's positive feedback on the novel data-centric perspective, remarkable performance, and potential of oracle query.
>
> **W1: What is the main difference between this work and existing works?**
>
> 1. **High-level Design and Data-Centric Perspective.** Our framework emphasizes a data-centric methodology, leveraging VLM and/or LLM primarily for data curation to address information asymmetry. Unlike prior research, which often uses these models to enhance retrieval performance directly, our work aims to underscore the critical role of data quality and the impact of data-driven strategies in this domain.
> 2. **Testing Phase Enrichment.** Existing work rarely consider the testing phase enrichment. We present a holistic solution of text query enrichment in testing phase, including query generation, query selection, and result aggregation.
> 3. **Dependency of LLM.** As we discussed in W1, the work does not have strong dependency on LLM. Using more cheap toolkits, e.g., NLTK can still harness the query selection and aggregation mechanism, boosting the performance.
> 4. **Novelty of the Query Selection Mechanism.** We emphasize that our Query Selection mechanism introduces a rarely explored area within the domain. This component is critical, as it significantly impacts performance and data efficiency.
> 5. **Analytical Insights Beyond the Method.** Beyond the technical contributions, we believe our analytical findings add substantial value to the field. For instance, the Oracle query concept demonstrates the substantial untapped potential of improved text representations, paving the way for future research in this area.
>
> **W2: More explanation about the oracle query would help understanding.**
>
> We thank the reviewer for raising this valuable question.
> In A.2. of the supplementary, we provide more details about how to select the oracle query. Moreover, we present statistics on the distribution of the oracle query. The result based on the MSR-VTT 1k test set shows that $59.5\%$ of the selected oracle queries come from the GT query set, while $40.5\%$ of oracle queries are from the enriched queries. This actually provides an explanation of the huge performance gap between baseline and oracle, suggesting that the text queries can be specially investigated to enhance the retrieval.
>
> In addition, in A.8. of the supplementary, we provide qualitative examples to demonstrate how the generated queries benefit text-video retrieval. These cases can help understand the role of text queries in a more straightforward way.
>
> **W3: The performance gain seems to main come from Text Enrichment in Retrieval, what about the Training part?**
>
> We thank the reviewer for raising this question.
> We would like to clarify that the effectiveness of Text Enrichment of Training are better unveiled with the help of Text Enrichment in Retrieval. Specifically, as shown in Table 5, by comparing Exp 1 and Exp 4, the $1.0\%$ performance gap seems to be marginal. However, when comparing Exp 5 and Exp 6, the performance gap becomes larger, i.e.,  $1.9\%$, validating the effectiveness of the training design. The difference is likely due to that the training emphasizes more diversity scene coverage and generalization, while the original test set cannot fully reflect this potential. The involvement of Text Enrichment in Retrieval can better unleash the potential of the stronger model. We thank the reviewer for pointing out this issue and we will explicitly discuss this point in the revised paper.
>
> **W4: The computation cost  should be investigated.**
>
> We thank the reviewer for pointing out this issue.
> In the supplementary, we have a dedicated investigation about the computation cost in a real-world scenario. In W2 of Reviewer Tkyv, we have a comparison regarding efficiency and latency across different data generation tools. We will include it with a more dedicated discussion into the revised paper.

---

> > ### Author Response · Authors · 2024-11-21
> >
> > **W5 and Q1: Implementation detail of majority voting and release of code & data.**
> >
> > Thanks for the question. Here we provide a pseudo code version of the implementation of majority voting. We plan to release the full code and data upon paper acceptance.
> > ```
> > function majority_voting_aggregation(similarity_matrices):
> >     Input: List of similarity matrices [num_query, 1, num_target]
> >     Output: Aggregated performance metrics
> >
> >     Initialize voted_sim_sort as an empty list
> >
> >     For each query:
> >         Initialize rank_list and sim_sort_list as empty
> >         For each similarity matrix:
> >             Get similarity scores for the query
> >             Sort scores in descending order and save to sim_sort_list
> >             Compute the query's rank and save to rank_list
> >
> >         Find the most frequent rank in rank_list
> >         Select the corresponding sorted similarities and add to voted_sim_sort
> >
> >     Concatenate results in voted_sim_sort, sort, and extract ranks
> >     Return computed metrics from the ranks
> > ```
> >
> > We thank the reviewer for reviewing our paper and providing valuable suggestion, which can definitely help this paper more clear and stronger. We will include these feedback into our revised paper. We sincerely hope the response can help resolve the reviewer's concerns. Thanks.

---

> > > ### Author Response · Authors · 2024-11-24
> > >
> > > Dear reviewer 8p6G,
> > >
> > > Thank you for reviewing our paper. We have addressed your concerns in our submitted response and provided a revised version of the manuscript. As the rebuttal period is coming to an end, we kindly request you to review our rebuttal and share any further comments. We greatly appreciate your valuable feedback!
> > >
> > > Best Regards

---

> > > > ### Comment · Reviewer_8p6G · 2024-11-25
> > > >
> > > > The authors addressed my concerns well, so I'm raising the score to "Accept".

---

> > > > > ### Author Response · Authors · 2024-11-25
> > > > >
> > > > > Dear reviewer 8p6G,
> > > > >
> > > > > We sincerely appreciate your efforts on reviewing our paper and raising the score! We will involve your suggestions into the revised paper to make it stronger. Thanks!
> > > > >
> > > > > Best,
> > > > >
> > > > > Authors of Submission862

---

### Official Review · Reviewer_38bZ · 2024-11-03

**Soundness:** 2
**Presentation:** 2
**Contribution:** 2
**Rating:** 6
**Confidence:** 5

**Summary:**

Due to the information asymmetry between video and text in text-video retrieval (TVR), this paper introduces a data-centric framework aimed at enriching textual representations to better align with the rich information contained in video content. However, despite efforts to leverage vision-language models (VLMs) and large language models (LLMs), the method lacks significant innovation.

**Strengths:**

S1.  This paper shows a method of utilizing VLMs and LLMs to enrich textual representations in training and inference stage.
S2. The data-centric method proposed by this paper maybe useful in practice, as this paper provides evidence of its effectiveness.
S3.  The writing is well-structured and clear, making the paper easy to follow.

**Weaknesses:**

W1. The method does not present substantial innovation. The improvement in model performance appears to be primarily attributed to the inherent capabilities of VLMs and LLMs in visual and textual understanding.
W2. While the paper introduces a query selection mechanism and designs a Farthest Query Sampling (FQS) algorithm, it would benefit from exploring additional query selection algorithms to further validate FQS’s effectiveness.
W3. The comparative experiments with the latest methods are somewhat lacking. Most of the related work cited in the experiments on state-of-the-art (SOTA) methods is limited to publications from 2022.

**Questions:**

Q1.  Could you include experiments comparing the model’s performance with more recent methods beyond those published in 2022? This would enhance the context of the model's strengths and limitations relative to current advancements.
Q2. How might the proposed framework perform in real-world, large-scale text-video retrieval systems？
Q3.  Could you clarify how this work distinguishes itself from existing methods beyond leveraging VLMs and LLMs? Are there any unique components or methodological innovations that specifically contribute to the improvements in performance?

Below are a few suggestions which could help the authors to refine a better version of this work.

1. This paper addressed the issue of information asymmetry in text-video retrieval from a data-centric perspective, therefore leveraging VLMs for event-level captioning and LLMs for query diversification. While this approach primarily combined existing methods to enhance textual representations, emphasizing the uniqueness of the model structure would better highlight its innovative contributions.

2. As the paper mentioned, relevant and diverse queries are expected to be retrieved. After initially constraining the relevance of the generated queries, FQS is applied to iteratively select queries by maximizing the minimum distance between them.
It may be worth considering adding relevance constraints to FQS, such as setting a minimum similarity threshold between query embeddings. Similarity metrics, such as dot product or cosine similarity, could be used for this purpose. Incorporating both distance and similarity into FQS might enhance its performance.
Additionally, FQS could be also compared with methods that minimize the minimum distance between selected queries to further demonstrate its ability to select diverse queries effectively.
Overall, designing alternative algorithms as comparisons would provide a stronger demonstration of FQS’s effectiveness.

3. In this paper, Table 1 presents the performance of the latest models. However, Tables 2 and 3 do not include these methods, such as Cap4Video. Including these comparisons would provide a more comprehensive evaluation.

---

> ### Author Response · Authors · 2024-11-21
>
> We sincerely thank the reviewer for reviewing our paper. We are glad to see the reviewer's positive feedback on the practical effectiveness of our method, and well structured presentation of the paper writing.
>
> **W1: The improvement in model performance appears to be primarily attributed to VLM and LLM.**
>
> Thanks for raising the question. We would like to first emphasize that the main focus of this paper is to explore and demonstrate the potential of enriching text data in text-video retrieval. The data-centric perspective and approach fundamentally differ our work from other model-centric methods, i.e., the main motivation is not transferring the capability of VLM and LLM into text-video retrieval.
>
> In fact, we have also explored a non-VLM, non-LLM version model. As shown in Table 6, NLTK is used as a tool for enriching the user query. We found that this "cheap" strategy can still boost a vanilla text-video retrieval model performance from 46.7 to 48.3.
>
> We acknowledge that foundation models, like VLM and LLM are powerful. However, how to effectively make use of them in text-video retrieval is also non-trivial. For example, although LLMs have strong text understanding capability, as decoder-only models, their feature embedding shows significant worse alignment than encoder models, as explored in recent work [1]
>
> Based on the above evidence, we believe that it might not be appropriate to simply attribute the performance gain to the utilization of foundation models.
>
> [1] Exploring the Role of Large Language Models in Prompt Encoding for Diffusion Models. NeurIPS 2024.
>
> **W2: It would benefit from exploring additional query selection algorithms to further validate FQS’s effectiveness.**
>
> Thanks for this valuable suggestion. Comparing the FQS with other possible query selection algorithms can definitely help make this paper stronger.
> In the current version, we have compared FQS with random selection as a baseline, and k-DPP algorithm. In rebuttal, we further involve nearest neighbor and farthest neighbor for comparison. The results shown in the Table below. It further demonstrates the superiority of the FQS algorithm. We will include this comparison into the revised paper.
>
> | Query Selection (k=2) | Rank-1 | Rank-5 | Rank-10 |
> |----------------------|--------|--------|---------|
> | Random               | 49.6   | 76.1   | 85.9    |
> | Nearest Neighbor     | 50.4   | 76.0   | 85.7    |
> | Farthest Neighbor    | 51.4   | 76.5   | 85.8    |
> | k-DPP                | 50.9   | 76.7   | 85.8    |
> | FQS                  | 52.1   | 76.8   | 86.3    |
>
> **W3 and Q1: Better to compare more recent works.**
>
> Thanks for the suggestion. We have updated our paper. In Table 1 of the main result, we include more recent works into comparison, including papers from ICCV 2023, CVPR 2024, ECCV 2024, and NeurIPS 2024. We would appreciate that if you could remind us of any missing works.

---

> > ### Author Response · Authors · 2024-11-21
> >
> > **Q2: How might the proposed framework perform in real-world, large-scale text-video retrieval systems?**
> >
> > Thanks for the interesting yet practical question.
> > The training part can increase the generalization capability. In real world, with large-scale datasets, the model can become stronger.  As for testing (deployment) in real-world scenario:
> > 1. High retrieval accuracy. As suggested by text-enrichment in retrieval phase, even without re-training the retrieval model, by only manipulating the text query, we can get a substantial performance gain.
> > 2. Efficiency. As we analyzed in the Appendix (Table 9), in real-world applications with large scale videos and queries, our method appears to be less efficient due to the involvement of LLM. However, this limitation can still be addressed. On the one hand, reducing inference cost of LLM is widely studied in industry, which can be transferred into this model. On the other hand, even without LLM, our model still shows decent performance with vanilla NLP toolkit, such as NLTK.
> >
> > We thank the reviewer for this constructive question. We will include a dedicated section to discuss the real-world application potential and challenges.
> >
> > **Q3: It is better to distinguish this work from existing works that also use VLM and/or LLM.**
> >
> > 1. **High-level Design and Data-Centric Perspective.** Our framework emphasizes a data-centric methodology, leveraging VLM and/or LLM primarily for data curation to address information asymmetry. Unlike prior research, which often uses these models to enhance retrieval performance directly, our work aims to underscore the critical role of data quality and the impact of data-driven strategies in this domain.
> > 2. **Testing Phase Enrichment.** Existing work rarely consider the testing phase enrichment. We present a holistic solution of text query enrichment in testing phase, including query generation, query selection, and result aggregation.
> > 3. **Dependency of LLM.** As we discussed in W1, the work does not have strong dependency on LLM. Using more cheap toolkits, e.g., NLTK can still harness the query selection and aggregation mechanism, boosting the performance.
> > 4. **Novelty of the Query Selection Mechanism.** We emphasize that our Query Selection mechanism introduces a rarely explored area within the domain. This component is critical, as it significantly impacts performance and data efficiency.
> > 5. **Analytical Insights Beyond the Method.** Beyond the technical contributions, we believe our analytical findings add substantial value to the field. For instance, the Oracle query concept demonstrates the substantial untapped potential of improved text representations, paving the way for future research in this area.
> >
> > We hope these clarifications address your concerns and enhance the overall understanding of our contributions. Thank you for the opportunity to improve our work.

---

> > > ### Author Response · Authors · 2024-11-24
> > >
> > > Dear reviewer 38bZ,
> > >
> > > Thank you for reviewing our paper. We have addressed your concerns in our submitted response and provided a revised version of the manuscript. As the rebuttal period is coming to an end, we kindly request you to review our rebuttal and share any further comments. We greatly appreciate your valuable feedback!
> > >
> > > Best Regards

---

> ### Comment · Reviewer_38bZ · 2024-11-25
>
> I have already raised the final ratings based on the authors' rebuttal. Besides, below are a few suggestions which could help the authors to refine a better version of this work (also upated in the previous comments).
>
> 1. This paper addressed the issue of information asymmetry in text-video retrieval from a data-centric perspective, therefore leveraging VLMs for event-level captioning and LLMs for query diversification. While this approach primarily combined existing methods to enhance textual representations, emphasizing the uniqueness of the model structure would better highlight its innovative contributions.
>
> 2. As the paper mentioned, relevant and diverse queries are expected to be retrieved. After initially constraining the relevance of the generated queries, FQS is applied to iteratively select queries by maximizing the minimum distance between them. It may be worth considering adding relevance constraints to FQS, such as setting a minimum similarity threshold between query embeddings. Similarity metrics, such as dot product or cosine similarity, could be used for this purpose. Incorporating both distance and similarity into FQS might enhance its performance. Additionally, FQS could be also compared with methods that minimize the minimum distance between selected queries to further demonstrate its ability to select diverse queries effectively. Overall, designing alternative algorithms as comparisons would provide a stronger demonstration of FQS’s effectiveness.
>
> 3. In this paper, Table 1 presents the performance of the latest models. However, Tables 2 and 3 do not include these methods, such as Cap4Video. Including these comparisons would provide a more comprehensive evaluation.

---

> > ### Author Response · Authors · 2024-11-25
> >
> > Dear reviewer 38bZ,
> >
> > We sincerely thank you for reviewing our paper, providing constructive suggestions, and raising the score.
> >
> > 1. We thank the reviewer for this valuable suggestion regarding the paper writing. We will further polish the paper to highlight the contribution on the overall model structure aspect.
> >
> > 2. We thank the reviewer for the constructive suggestion. Designing alternative algorithms as comparisons can definitely better demonstrate the effectiveness of FQS algorithm. We show the extended experiment results in the Table below.
> >
> >
> > | Query Selection (k=2) | Rank-1 | Rank-5 | Rank-10 |
> > |----------------------|--------|--------|---------|
> > | Random               | 49.6   | 76.1   | 85.9    |
> > | Nearest Neighbor     | 50.4   | 76.0   | 85.7    |
> > | Farthest Neighbor    | 51.4   | 76.5   | 85.8    |
> > | k-DPP                | 50.9   | 76.7   | 85.8    |
> > | FQS (s=0.5)          | 52.1   | 76.8   | 86.3    |
> > | FQS (s=0.75)         | 52.1   | 76.8   | 86.3    |
> > | FQS (s=0.85)         | 51.6   | 76.7   | 86.2    |
> > | FQS (s=0.95)         | 49.8   | 76.1   | 86.3    |
> > | NQS                  | 49.9   | 75.8   | 86.1    |
> > | FQS                  | 52.1   | 76.8   | 86.3    |
> >
> > Firstly, we experiment with the suggested **"minimum similarity threshold"**, denoted as FQS (s=) in the Table, where the value of "s" is the similarity threshold. In implementation, when iteratively selecting queries, we add an additional condition that requires the selected queries to meet the minimum similarity. If no query meets this condition, we will use the original user query. The results show that when setting the threshold to 0.5 or 0.75, the performance is the same as without the condition, suggesting that most of the queries are distributed in a closely relevant range. When further increasing the threshold to 0.85 or 0.95, we find that the retrieval performance starts to decrease. It is due to that higher similarity threshold will limit the diversity of the selected queries and constrain the retrieval performance.
> >
> > Secondly, we implement the suggested **"minimize the minimum distance"** version, which is denoted as NQS (Nearest Query Sampling) in the Table.
> > Evaluation results show that this approach will also restrict the diversity of the selected queries, thereby hurting the retrieval performance.
> >
> > These experiments can further verify the effectiveness of the proposed FQS algorithm. We greatly thank the reviewer for this valuable and constructive feedback. We will include this experiment in the cam-ready paper.
> >
> > 3. Thanks for the kind reminder. We have updated the paper to include more comparisons in Table 2 and Table 3.
> >
> > In the cam-ready version, we will further involve your suggestions to make the paper stronger. Thanks!
> >
> > Best,
> >
> > Authors of Submission862

---

### Official Review · Reviewer_Tkyv · 2024-11-03

**Soundness:** 2
**Presentation:** 3
**Contribution:** 2
**Rating:** 5
**Confidence:** 5

**Summary:**

This work aims to solve the problem of information asymmetry between text descriptions and videos in text-video retrieval,i.e., text often contains only part of the video content. The author uses VLM to generate more captions from videos during training; and uses LLM to expand queries and add more content during inference.

**Strengths:**

1. The proposed problem of information asymmetry in the text-video retrieval is important and needs to be solved in the development of this task.
2. The paper is well written and easy to follow.

**Weaknesses:**

1. The proposed method uses VLM and LLM to generate details to the original caption. The captions and queries generated in this way are unreliable and contain a lot of information that is not in the video. In the inference phase, without the addition of oracle information, the newly generated captions and querys are hallucinated by the large language model. The authors need to provide more evidence to support their idea.

2. The work involves utilizing large visual language models and large language models. The required training cost and inference computation are much higher than compared works. It`s not fair to compare directly in experiments. The authors need to prove the performance gain is not from the accumulation of more computation.

3. Line 238 mentions that this work used an image captioner. The designed method does not consider the temporal cues in the video neither. The proposed method is essentially designed for an image-text retrieval task rather than a video-text retrieval task.

4. In line 278, the fact that oracle query can improve model performance does not mean that supplementing the query is useful. Introducing the ground truth query can greatly enhance model performance, regardless of whether the query is supplemented or not.

**Questions:**

1. Line 290 proposed to ensure the diversity in the supplemented queries. Why are the supplemented queries should be expressed in as many ways?

2. To verify the effectiveness of the proposed method, the authors should provide some examples of the generated captions during training and the expanded queries during inference. It can help to understand the contribution of the work.

3. The approach proposed in Figure 3 requires more illustration on design idea and details; the current version is a bit confusing.

4. The authors need to explain the training efficiency and inference efficiency of the proposed method.

Other questions seen in the Weaknesses.

---

> ### Author Response · Authors · 2024-11-21
>
> We sincerely thank the reviewer for reviewing our paper. We are glad to see that the reviewer and us have reached a consensus that **information asymmetry** is an important yet under-explored problem in text-video retrieval field. Our work, as a pioneering exploration on this problem from a data-centric perspective, aims to inspire more future research about this problem.
>
> Before diving into the response for specific questions, we would like to make a global clarification. As we noticed that some questions are possibly raised from misunderstanding of some concepts of the paper.
> In our work, we have the following types of text queries:
>
> - **User query**, also termed as "**ground-truth query**" in the paper, as it comes from a specific evaluation dataset. We use the two terms interchangeably in the paper.
> - **Enriched/Generated queries**, usually more than one text query, which are generated by the proposed text enrichment framework.
> - **Oracle query**, a single text query selected from the union set of above concepts, which yields the best retrieval result. Identifying the oracle query requires access to the video-text pair labels information, which is impractical in real application. Thus, the oracle query is **only used for analysis** to demonstrate the huge potential of text data. We do not use this label information in evaluation.
>
> The term of **ground-truth query** may mislead the readers to think that it also requires access to the label information, which does not align with the actual setting. We sincerely apologize for the misleading concepts. We will use **user query** in this rebuttal and in the revised paper for better clarity.

---

> > ### Author Response · Authors · 2024-11-21
> >
> > **W1: The captions and queries are generated by VLM and LLM, which may contain hallucination information. It is better to provide evidence on the reliability of using these models.**
> >
> > - Thanks for this constructive question. Indeed, VLM/LLM generated contents inevitably contain hallucinatory information. We have the following designs to address this potential issue.
> > 	- **Text-enrichment in training.** In this phase, the data are mainly generated by VLM via captioning. We use these data to "pre-train" the model. The model are later trained on the standard dataset with human annotated captions. This two-stage design makes the training procedure more robust to the noise in stage-1, as stage-2 can help rectify the noisy part with more reliable data.
> > 	- **Text-enrichment in testing.** In this phase, the data are mainly generated by LLM. To ensure the potential hallucination not harm the retrieval performance, we have three designs: 1) a carefully curated prompt that instructs the LLM to follow the semantic meaning of the user query; 2) a novel query selection mechanism that selects reliable yet diverse queries from the generated ones, its effectiveness is validated in the ablation study (Table 7); 3) a majority voting aggregation strategy that filters out the unreliable results. The three designs establish a hierarchy to remove hallucination while preserving useful data.
> > - Apart from the designs, we have the following evidences to prove this aspect.
> > 	- First, the **overall performance** gain of our method has effectively proved that these generated data can be used to significantly improve the performance, while the potential hallucination can be blocked or removed.
> > 	- The **choice of LLM** plays an important role in this issue, as stronger LLMs tend to generated more reliable contents, while weaker LLMs are more likely to generate hallucination. In Sec. 4.2.4 and Table 8, we compare a strong LLM GPT-4 and a weak LLM Phi-3.5, the result shows that 1) Phi-3.5 indeed generates more hallucinatory content than GPT-4; 2) Our method is robust to the choice of LLM, using Phi-3.5 can still achieve comparable performance to GPT-4. The result further indicates that our method is robust to hallucination.
> >
> > **W2: It is better to prove the performance gain is not from the accumulation of more computation, as VLM and LLM are involved in the proposed method.**
> >
> > - We thank the reviewer for pointing out this problem. We acknowledge that computation cost is an important aspect for a model, especially for model-centric methods with novel architecture designs. However, we would like to emphasize that our method, as a data-centric approach, focus more on the effect of data. Both VLM and LLM are simply used as data generation tools.
> > - Specifically, for VLM, it is used to generate **training data**, which is already an affordable choice compared to human annotation. The cost of such data generation are rarely taken into consideration of computation cost [1], as training is a one-pass process.
> > - For LLM, it is used to enrich text queries during testing phase. We agree that this needs to be investigated, since the LLM computation is used at each time of inference. We thank the reviewer for raising this valuable question. We analyze the computation and inference time introduced by the data text enrichment of test phase in the table below.
> >
> > | Text Enrichment Tool | Inference Time | GPU Usage | GPU Compute Increase | Time Increase | Overall Accuracy |
> > |---------|--------|-----|----------|-------|--------|
> > | None      | 16.55s | 1474MB    | -  | - | 47.7 |
> > | GPT-4 API  | 21.8s          | Unknown   | Unknown  | 31.72%  | 52.1 (+4.4)      |
> > | Phi-3.5 Locally Deployed | 29.96s         | 4560MB    | 2X                   | 81.02%        | 51.5 (+3.8)      |
> > | NLTK Toolkit | 16.71s | 1474MB    | 0 | 0.97%         | 50.5 (+2.8)      |
> >
> > In this table, we use 1k videos from the MSR-VTT dataset as the test set and measure the computation cost and time efficiency for using one user query to retrieve the target video. We observe that:
> >
> > - For each user query, involving LLM as the data generation tool can significantly boost the performance, but also increase the compute and time, ranging from 31.72% to 81.02%.
> > - Although GPT-4 is supposed to be a much larger model than Phi-3.5, its API time latency is much smaller. This gap reveals that there is large room for optimizing the time latency of Phi-3.5 in practical applications.
> > - As our main target is to investigate and reveal the potential of data, but not the data generation tool itself, we also explore using NLTK toolkit as a (almost) free lunch to generate data. It introduces negligible compute and time latency, showing a considerable performance gain compared to baseline, while still worse than LLMs.
> >
> > We thank the reviewer for raising the valuable suggestion. We will include the discussion into the revised paper.
> >
> > [1] Cap4Video: What Can Auxiliary Captions Do for Text-Video Retrieval?

---

> > > ### Author Response · Authors · 2024-11-21
> > >
> > > **W3: The usage of image captioner might not be appropriate for video tasks, as it lacks of awareness of temporal.**
> > >
> > > We thank the reviewer for raising this problem. We believe the usage of image captioner does not invalidate our method due to the follow aspects:
> > > - We **do not** use an image captioner model for the entire video. Instead, before image captioner, there is a Video Temporal Segmentation model, which is dedicated to managing video temporal dynamics.
> > > - After video temporal segmentation, each video clip only contain a short time window with only **a single scene**. As we discussed in the paper (Section 3.2.1), the short clip is regarded as an atomic scene according to our empirical study, and an image captioner is sufficient to capture the essential information.
> > > - The combination of video temporal segmentation and image captioning model considers both temporal dynamics and spatial semantics.
> > >
> > > **W4: Ground truth query can greatly enhance model performance, regardless of whether the oracle query is supplemented or not.**
> > >
> > > We thank the reviewer for pointing this valuable question, which is rarely discussed in the paper. We feel like this concern might be related to the concepts mentioned in the global clarification. We hope the clarification can make us on the same page.
> > > - **Effect of oracle query**.  The main purpose of introducing "Oracle query" is to demonstrate the huge potential of manipulating text query based on an existing text-video retrieval model. This analysis serves as a basis of data-centric research in text-video retrieval. Ideally, the ultimate goal is to design an algorithm that can find the **best query** among a set of candidates. We hope this analysis can inspire future research. We agree with the reviewer that this result does not mean supplementing the oracle query is helpful. That's very true, as the supplemented query need to work with the existing queries together to reach a consensus. We greatly thank the reviewer for providing such a valuable suggestion. We will discuss this point more explicitly in the revised paper.
> > > - **The role of ground-truth query.** We would like to clarify that the ground-truth query is the same comcept as **user query**, which is necessary in any retrieval system. It should be regarded a baseline, but not "greatly enhance model performance". We do not include any other "ground-truth" information during evaluation.
> > >
> > >
> > > **Q1: Why are the supplemented queries should be expressed in as many ways?**
> > >
> > > Thanks for the question. This originates from the essential problem of **information asymmetry** between video and text, as videos contain much more richer information than texts. Therefore, we expect the text queries can be expressed in a more diverse ways, so that it can complement the asymmetry and increase the successful retrieval rate.
> > >
> > > **Q2: Providing more examples of the generated text queries of training and testing can help better understand the contribution of this work.**
> > >
> > > We thank the reviewer for this constructive suggestion. In the Appendix (A. 8), we provide qualitative examples of the inference text queries and analyze how the enriched queries benefit retrieval. Following the reviewer's suggestion, we further provide more examples of enrichend data of training phase in Appendix in the updated paper.
> > >
> > > **Q3: It is better to further polish Figure 3 to make it more clear.**
> > >
> > > Thanks for this valuable suggestion. We made a new version of this figure to help understand the process, which is updated in the paper. We will further polish the figure and text to improve the paper.
> > >
> > > **Q4: The authors need to explain the training efficiency and inference efficiency of the proposed method.**
> > >
> > > Thanks for the suggestion. The inference efficiency is analyzed in the response of W2. As for the training efficiency, it depends on the amount of the generated data. In the implementation of our paper, we generate captions around 5 times of the original text captions. Therefore, the training cost has increased 5X. Note that we also compare our model to other models with the same training cost for a fair comparison, as shown in Table 5 of the paper. We thank the reviewer for pointing out this problem and we will have a dedicated discussion  in the revised paper.
> > >
> > > We thank the reviewer for reviewing our paper and providing valuable suggestion, which can definitely help this paper more clear and stronger. We will include these feedback into our revised paper. We sincerely hope the response can help resolve the reviewer's concerns. Thanks.

---

> > > > ### Author Response · Authors · 2024-11-24
> > > >
> > > > Dear reviewer Tkyv,
> > > >
> > > > Thank you for reviewing our paper. We have addressed your concerns in our submitted response and provided a revised version of the manuscript. As the rebuttal period is coming to an end, we kindly request you to review our rebuttal and share any further comments. We greatly appreciate your valuable feedback!
> > > >
> > > > Best Regards

---

> ### Comment · Reviewer_Tkyv · 2024-11-25
>
> The authors have addressed my concerns regarding efficiency, the concept of oracle, as well as the writing and figures in the paper. However, I still have concerns about the paper's core novelty, supplementing captions by VLMs and LLMs. Although the authors have proposed some methods to mitigate hallucinations, this method inevitably introduces the hallucination issue of language models. I have raised my score to 5.

---

> > ### Author Response · Authors · 2024-11-25
> >
> > Dear reviewer Tkyv,
> >
> > We sincerely appreciate your efforts on reviewing our paper and raising the score.
> >
> > Regarding your concern about hallucination of VLMs and LLMs, existing experiment results show that our method is robust to such noisy data. We acknowledge that hallucination is still a potential limitation that may affect the performance. In the updated paper, we have added a discussion about this limitation in the Appendix.
> >
> > In the cam-ready version, we will further involve your suggestions to make the paper stronger. Thanks.
> >
> > Best,
> >
> > Authors of Submission862

---

### Meta-Review · Area_Chair_kmFn · 2024-12-19

**Metareview:**

This paper tackles the challenge of text-video retrieval. To enhance text-visual alignment, it introduces a method that leverages Vision-Language Models (VLMs) for event-level captioning and Large Language Models (LLMs) for query diversification, alongside a query selection mechanism. The strengths of the paper include the importance and usefulness of the task, clear writing, and good performance demonstrated on multiple benchmarks.

Several weaknesses were addressed during the rebuttal phase. Ultimately, two out of three reviewers expressed support for accepting the paper. The remaining concern, raised by the opposing reviewer, centers on the potential for hallucinations introduced by language models. In the AC's opinion, it does not appear to be a substantial weakness. The paper’s empirical validation demonstrates effectiveness, which has been recognized by other reviewers. In addition, the authors have acknowledged this issue in the rebuttal and committed to including a discussion in the revised paper.

**Additional Comments On Reviewer Discussion:**

All three reviewers engaged in the rebuttal discussion, and all raised their ratings afterward.

---

### Decision · Program_Chairs · 2025-01-22

Accept (Poster)